# Polyunsaturated Phospholipids Increase Cell Resilience to Mechanical Constraints

**DOI:** 10.3390/cells10040937

**Published:** 2021-04-17

**Authors:** Linette Kadri, Amélie Bacle, Spiro Khoury, Clarisse Vandebrouck, Jocelyn Bescond, Jean-François Faivre, Thierry Ferreira, Stéphane Sebille

**Affiliations:** “Lipotoxicity and Channelopathies (LitCh)-ConicMeds” Laboratory, University of Poitiers, 86000 Poitiers, France; linette.kadri@univ-poitiers.fr (L.K.); amelie.bacle@gmail.com (A.B.); spiro.khoury@univ-poitiers.fr (S.K.); clarisse.vandebrouck@univ-poitiers.fr (C.V.); jocelyn.bescond@univ-poitiers.fr (J.B.); jean-francois.faivre@univ-poitiers.fr (J.-F.F.); stephane.sebille@univ-poitiers.fr (S.S.)

**Keywords:** polyunsaturated fatty acids, membrane plasticity, muscle cells, mechanical constraints, docohexaenoic acid (DHA)

## Abstract

If polyunsaturated fatty acids (PUFAs) are generally accepted to be good for health, the mechanisms of their bona fide benefits still remain elusive. Membrane phospholipids (PLs) of the cardiovascular system and skeletal muscles are particularly enriched in PUFAs. The fatty acid composition of PLs is known to regulate crucial membrane properties, including elasticity and plasticity. Since muscle cells undergo repeated cycles of elongation and relaxation, we postulated in the present study that PUFA-containing PLs could be central players for muscle cell adaptation to mechanical constraints. By a combination of in cellulo and in silico approaches, we show that PUFAs, and particularly the ω-3 docosahexaenoic acid (DHA), regulate important properties of the plasma membrane that improve muscle cell resilience to mechanical constraints. Thanks to their unique property to contortionate within the bilayer plane, they facilitate the formation of vacuole-like dilation (VLD), which, in turn, avoid cell breakage under mechanical constraints.

## 1. Introduction

Although it is generally accepted that polyunsaturated fatty acids (PUFAs), especially ω-3 fatty acids (FA), are good for health, the underlying mechanisms still remain elusive [1]. PUFAs exert their physiological actions through three different forms: as free molecules, as precursors of bioactive molecules, or as components of membrane phospholipids (PLs) [1,2].

PLs, which bear two FA chains at the sn-1 and sn-2 positions of their glycerol backbone, are the main components of cellular membranes. In mammalian cells, phosphatidylcholine (PC) and phosphatidylethanolamine (PE) are the most abundant PLs [3]. These species, and to a lesser extent, phosphatidylinositol (PI) and phosphatidylserine (PS), constitute the main lipid classes whatever the organ considered, constituting 75% of all lipid species in the heart of rats, and 79% in the liver, as examples [3]. The FA diversity in phospholipids applies mostly to the sn-2 position of the glycerol backbone, resulting in asymmetric phospholipids containing a saturated FA at position sn-1 and an unsaturated FA at position sn-2 [4,5].

Interestingly, organs display very characteristic fatty acid profiles within their phospholipids [6,7,8]. This results from the activity of acyltransferases, which selectively incorporate defined fatty acids into PLs [7,9]. A very simple way to obtain a rapid overview of this fatty acid distribution is to consider the double bond (DB) index of a given lipid class, which corresponds to the relative percentage of saturated (DB = 0: i.e., PLs containing no double bond within their FA chains) versus monounsaturated (DB = 1: one double bond), diunsaturated (DB = 2: two double bonds) and polyunsaturated (DB > 2: >two double bonds) species within this lipid class. In rats, PC is the lipid class that displays the highest diversity in terms of its fatty acyl chain composition depending on the organ considered. When considering PC, rat organs display very characteristic fatty acid distributions within PLs, presenting various saturation rates: they can be classified as DB = 0 (spleen, lung), DB = 1 (brain), DB = 2 (pancreas) and DB > 2 organs (liver, muscle, and cardiovascular system) [6,7,8].

Another intriguing observation is that, among PUFAs, very specific signatures can be observed among organs when considering the relative percentages of ω-3 versus ω-6 PUFAs [6,7,8]. The main PUFA species found within PLs are the ω-6 arachidonic acid (AA; 20:4ω-6) and the ω-3 docosahexaenoic acid (DHA; 22:6ω-3). In rats, one can differentiate AA- (liver and cardiovascular system) and DHA-enriched (skeletal muscle) organs [6].

Finally, the various organs do not respond similarly to selective diets [6,10,11]. Saturated and monounsaturated FAs originating from a high-fat diet preferentially accumulate in rats within PC in their liver and skeletal muscles, in a process that occurs at the expense of PUFA. Interestingly, this FA redistribution is paralleled by a global decrease in muscle strength [6].

A still open question is the physiological meaning of such enrichment in DHA in muscle PLs. Recent experiments and simulations suggest that PUFA-containing PLs respond differently to mechanical stress than other PLs, owing to their unique conformational plasticity [12,13]. Mechanoprotection is extremely important for the integrity of tissues undergoing constant mechanical stress, such as muscle tissue. Since muscle cells undergo repeated cycles of elongation and relaxation and are constantly exposed to mechanical stress, we postulated in the present study that PUFA-, and more specifically DHA-containing PLs may be central players for muscle cell adaptation to mechanical constraints.

With this aim, we first determined fatty acid distribution within PLs in different mouse organs and reproduced the characteristic signature observed in muscle in vivo, in C2C12 muscular cells in vitro. In a second step, we evaluated the impacts of such a signature on C2C12 response to mechanical constraints under the form of an osmotic downshock. We show that PUFAs, and particularly DHA, regulate important properties of the plasma membrane that improve the cell resilience to mechanical constraints. These observations suggest that DHA enrichment in muscle tissues may not be haphazard but the result of muscle adaptation to imposed mechanical constraints.

## 2. Materials and Methods

### 2.1. Animals

The present study was approved by the Comité d’Ethique et d’Expérimentation Animale (COMETHEA) and the French Ministère de l’Enseignement Supérieur, de la Recherche et de l’Innovation (authorization n°2016071215184098). The protocols were designed according to the Guiding Principles in the Care and Use of Animals approved by the Council of the American Physiological Society and were in adherence with the Guide for the Care and Use of Laboratory Animals published by the US National Institutes of Health (NIH Publication no. 85-23, revised 1996) and according to the European Parliament Directive 2010/63 EU.

All animal handling and experiments were performed at the PreBios facility (University of Poitiers, France). Organs were isolated from mice euthanized by lethal injection. Male C57BL10 mice (Janvier, Le Genest-Saint Isle, France) were housed in a temperature-controlled room in a 12 h light/dark cycle environment with ad libitum access to water and food (standard diet). On the day of the experiment, mice were anesthetized with an i.p. injection of 0.8 mL urethane 10% (*n* = 8). Organs selected for lipid analysis (brain, liver, tibialis anterior skeletal muscle, pancreas, lung, right cardiac atria, and right ventricle) were quickly excised and put on ice, after which they were cut into small pieces (2 mm^3^) and dipped into liquid nitrogen. The frozen pieces were introduced into cryotubes before immersion in liquid nitrogen for storage at −80 °C.

### 2.2. Cell Culture

C2C12 (ATCC^®^ CRL-1772™; ATTC, Manassas, VA, USA) mouse skeletal myoblasts were grown at 37 °C with a 5% CO_2_ humidified atmosphere in Dulbecco’s Modified Eagle’s Medium with 4.5 g L^−1^ Glucose (DMEM; Lonza Bioscience, Bâle, Zwitzerland), supplemented with 10% fetal bovine serum (FBS; Lonza Bioscience, Bâle, Zwitzerland).

For fluorescence imaging, C2C12 Myoblasts were directly seeded in 35 mm-diameter tissue culture dishes with a bottom made from a glass coverslip coated with Matrigel (Corning, Corning, NY, USA).

For lipid analyses, approximately 10^6^ cells grown under the various conditions were harvested, washed with Dulbecco’s Phosphate Buffered Saline (DPBS; Lonza Bioscience, Bâle, Switzerland), and the cell pellets were stored at −20 °C until lipid extraction.

### 2.3. Fatty Acid Preparations and Cell Treatment

Linoleic Acid (18:2ω-6), linolenic acid (18:3ω-3), AA, and DHA (Santa Cruz Biotechnology, Dallas, TX, USA), were prepared in absolute ethanol at 100 mM. Each of these stock solutions was directly diluted to a C2C12 culture medium, at 100 µM final concentration. Cells were incubated at 37 °C, for 16 h, under these conditions.

### 2.4. Lipid Extraction, Phospholipid Purification, and Mass Spectrometry Analyses

Lipids were extracted from each individual sample (C2C12 cells or organ samples), according to the following procedure. Each frozen sample was first submitted to three rounds of grinding using a Precellys Evolution homogenizer (Bertin Technologies, Montigny-le-Bretonneux, France) and resuspended into 1 mL of water before transfer into glass tubes containing 500 μL of glass beads (diameter 0.3–0.4 mm; Sigma-Aldrich, Saint-Louis, MO, USA). Lipids were extracted using chloroform/methanol (2:1, *v*/*v*) and shaking with an orbital shaker (IKAH VXR basic VibraxH—Sigma-Aldrich, Saint-Louis, MO, USA) at 1500 rpm for at least 1 h, as already described elsewhere [14]. The final organic phase was evaporated and dissolved in 100 μL dichloromethane for purification of PLs on a silica column (Bond ELUT-SI—Agilent Technologies, Santa Clara, CA, USA). Lipid samples were loaded on the top of the column. Non-polar lipids were eluted by the addition of 2 mL dichloromethane and glycolipids with 3 mL acetone. PLs were then eluted by 2 mL chloroform/methanol/H_2_O (50:45:5, *v*/*v*/*v*).

PL analysis in mass spectrometry (MS) was performed by a direct infusion of purified lipid extracts on a Synapt G2 HDMS (Waters Corporation, Milford, CT, USA) equipped with an electrospray ionization source (ESI). The mass spectrum of each sample was acquired in the profile mode over 1 min. The scan range for PL analysis was from 500 to 1200 *m*/*z*. PC species were analyzed in positive ion mode after the addition of 0.1% (*v*/*v*) formic acid as already described [14]. PE, PI and PS species were analyzed in negative ion mode after the addition of 0.1% (*v*/*v*) triethylamine. Identification of the various PL species was based on their exact mass using the ALEX (Analyses of Lipid Experiments) pipeline [15]. Tandem mass spectrometry experiments (MS/MS) were also carried out when structural information about phospholipid species composition is needed. These MS/MS experiments were performed by collision-induced dissociation (CID) mode, as described elsewhere [6].

### 2.5. Cell Labeling for Confocal Microscopy and Methods for Image Analyses

C2C12 cells were seeded 35 mm-diameter tissue culture dishes with a bottom made from a glass coverslip coated with Matrigel (Corning, Corning, NY, USA) and treated with or without AA or DHA during 16 h, at 37 °C, as described above. The medium was removed, and each dish was washed three times with cold DPBS buffer without CaCl_2_ (Lonza Bioscience, Bâle, Switzerland). Then, each dish was incubated with the FM1-43 probe (*N*-(3-Triethylammoniumpropyl)-4-(4-(Dibutylamino) Styryl) Pyridinium Dibromide), a water-soluble dye, which is nontoxic to cells and virtually nonfluorescent in an aqueous medium, and inserts into the outer leaflet of the cell membrane where it becomes intensely fluorescent at 515 nm [16,17]. The working staining solution at 5 µg.mL^−1^ final concentration was prepared in ice-cold DPBS without calcium from 100 µg.mL^−1^ stock. Each dish was incubated on ice, with 1 mL of working solution for 5 min, in the dark. Dishes were mounted on a microscope and imaged in staining solution without washing. Finally, the osmotic shock was processed by adding 1 mL of hypotonic solution, as described elsewhere (33), directly in the dishes. The hypotonic solution contained (in mM): 70 NaCl, 5.4 KCl, 1.8 CaCl_2_, 1.1 MgCl_2_, 5 HEPES, and 10 glucose, pH 7.4. All experiments were carried out at room temperature.

Living cells stained with FM1-43 were then examined with a laser scanning confocal microscope (Olympus FV1000, Olympus, Tokyo, Japan). A 405 nm diode was used for excitation of FM1-43, and double emission fluorescence was recorded simultaneously through two spectral detection channels at 515 nm (green fluorescence emission) and 453 nm (yellow fluorescence emission). The 640 × 640 pixels images were acquired with UPLSAPO 60X W NA:1.20 objective lens. Numerical zooming to 0.13 µm pixel size was conducted with respect to oversampling Nyquist criteria. A 3D optical sectioning of 0.2 to 0.4 µm was driven with a step *Z*-axis motor. The 3D images were analyzed with Imaris software (Bitplane, Belfast, UK). Time sequences of both transmission and fluorescence confocal images have been recorded in FM1-43 loaded cells. Hence, in each sequence, 300 fluorescence images with 5 s time resolution (time between two fluorescence acquisitions for each pixel position) were recorded on the same microscope field in cell preparations. Sequences of images were analyzed with ImageJ allowing us to follow fluorescence in each region of interest as a function of time.

To monitor the formation of vacuole-like dilations (VLDs), mean fluorescence was counted in circle regions of interest surrounding a given VLD, normalized to basal fluorescence, and reported as a function of time. Fluorescence kinetics were analyzed with a computer program developed in our laboratory under IDL 5.3 structured language on a PC computer. Several parameters were extracted from the analyzed curves: amplitude (Plateau) and kinetic parameters of VLD formation (t0: time to fluorescence appearance, V_max_: maximal rate of fluorescence increase, and t_Vmax_: time to V_max_).

Surface variations were determined by manually measuring the surface of individual cells in a field as a function of time using transmission images (non-treated (NT) VLD, *n* = 47; NT cracked, *n* = 58; AA, *n* = 96; DHA, *n* = 88; four independent experiments). Since myoblasts can be considered as hemispheres, variations in the area of a given cell, provided that the distance between this given section and the base of the hemisphere remains constant (which is guaranteed by the fact that cells are adhered to the support), are directly proportional to surface variations of the overall hemisphere. The initial surface was determined before the osmotic downshock. When the maximum surface was reached following this treatment (surface max), the surface of the cross-section was determined, and the corresponding time was noted (t_Surface max_). The % increase in the surface was determined from these initial and maximum surface values.

To evaluate plasma membrane rupture and cell breakage, the fluorescence of FM1-43 at 543 nm was determined. Indeed, under cell breakage, FM1-43 rapidly fills the cytoplasm and generates an intense fluorescence at this wavelength [18]. The mean fluorescence at 543 nm was measured in C2C12 cells as a function of time after induction of the osmotic downshock. At the end of this osmotic stress, the fluorescence intensity, relative to basal (relative fluorescence intensity), was determined for each cell and expressed as arbitrary units (A. U.). Cell breakage was determined by the combination of transmission observation and increase in cytoplasmic fluorescence at 543 nm. Using this method, we could determine that relative fluorescence intensities above 10 corresponded to bona fine cell breakage. When cell breakage was visualized, the time when this breakage occurred was also determined (t breakage).

### 2.6. Molecular Dynamics Simulations

All molecular dynamics (MD) simulations the MARTINI coarse-grain (CG) model [19,20]. The systems were built using the insane script [21] with the defined lipid parameters (with four beads for the oleoyl tail).

The systems contained only PC species since they are the main components of human cellular membranes. The composition of the systems was optimized to fit at best the lipidomic results from MS. To do so, we mixed four different types of fatty acid tails (palmitoyl, 16:0; oleoyl, 18:1; AA, 20:4; and DHA, 22:6) in the following combinations: 16:0-16:0, 16:0-18:1, 18:1-18:1, 16:0-20:4, and 16:0-22:6. Note that the coarse-grained simplification does not distinguish C16:0 from C18:0, C20:4 from C20:5, and C22:6 from C22:5. The two leaflets were equivalent, and no flip-flop events were observed. Every system was composed of around 3000 lipids at full hydration with 10 standard MARTINI water molecules [19] per lipid. The membrane composition was obtained by mixing the selective PC species indicated in the relevant Figures at a 50%/50% ratio.

All simulations were performed using the GROMACS simulation package version 4.5.3 [22], using an integration time step of 30 fs. The temperature was maintained at 310 K (37 °C) using the v-rescale algorithm [23] with a relaxation time of 1.0 ps. Semi-isotropic pressure coupling was applied with a reference pressure of 1 bar using the Parrinello–Rahman algorithm [24,25] with a relaxation time of 12.0 ps; van der Waals interactions were turned off between 0.9 and 1.2 nm using the shift function in GROMACS. This shift function was also used to turn off the electrostatic interactions between 0 and 1.2 nm, and a dielectric constant of 15 was used.

To mimic a membrane extension, we performed several simulations in an NPaT condition by constraining the X and Y value of the box (membrane plane) to remain constant during the simulation. The area per lipid of each system at equilibration was used as a reference value. We first increased the X and Y values of the membrane plane by 5%. We set the area of the membrane constant by reducing the compressibility of the barostat at 0 bar-1. We performed simulations of 210 ns and replicated them three times for each system. To obtain the next system, we took the last image of the 5% extended system and increased the X/Y values again and so on until persistent breakage of the systems. Using this method, we were able to perform extensions from 5% up to 25%.

Surface tension was determined as already described elsewhere [26]. For all simulations, surface tension (γ) was computed from the diagonal values of the pressure tensor (*P_xx_*, *P_yy_*, and *P_zz_*) using the Kirkwood–Irving method [27]: because of high fluctuations in *P_xx_*, *P_yy_*, and *P_zz_*, γ fluctuates vigorously on a microscopic system of a few thousands of atoms such as ours. Thus, two simulations of the same system with constant xy total area but with different initial velocities can lead to different values of g on the hundreds of ns timescale. For this reason, the average and error on γ were evaluated from the g_energy GROMACS tool (g_energy outputs to the screen average and error using points at all time steps, which is more precise than generating an xvg file and doing the analysis on the latter) using the whole trajectory for each simulation. In this manuscript, we report a single value, in nN m^−1^, for each system, which was obtained by concatenating all trajectories of a given system prior to the procedure with g_energy.

The acyl chain angle was computed as the angle between two vectors. The first vector is defined by the first and last bead of the acyl chain. The second vector is the perpendicular axis to the membrane plane. An angle close to 0° means that the acyl chain of the lipid is perpendicular to the membrane plane, whereas when the angle is close to 90°, the acyl chain is in the membrane plane. We then represent these angles as probabilities for each extension value applied to the systems.

### 2.7. Statistical Analyses

*p* values were calculated either by two-tailed *t*-tests or ANOVA, completed by adequate post-tests, as indicated in the corresponding figure legends. These analyses were performed using the GraphPad Prism 5 software. ns: non-significant; **** *p* < 0.0001, *** *p* < 0.001, ** *p* < 0.01 and * *p* < 0.05. The multivariate analysis of the dataset, represented by principal component analysis (PCA), was performed using the Past software version 3.25 (http://folk.uio.no/ohammer/past, accessed on 6 February 2020). For the PCA analysis, the double bond (DB) index of a given lipid was used to compute the PCA. When considering the dataset corresponding to phosphatidylcholine species, PCA-1 explained 62.8% of the variation in the data, and PCA-2 explained 32.7% of the residual variation.

## 3. Results

### 3.1. Fatty Acid Distribution within Phospholipids Varies Depending on the Organ

In a first step, we determined the fatty acid distribution within the different PLs species, namely PC, PE, PS, and PI, in various organs from mice under a standard diet. The present study was performed on representative digestive organs (liver and pancreas), skeletal muscles (tibialis anterior muscle), the heart (right ventricle and atrium), the brain, and the lung. With this aim, total PLs were extracted from the various organs and analyzed by mass spectrometry in both the positive and negative ion modes, as described in the previous section. The results obtained for PC are displayed in Figure 1 as radar graphs and Appendix A as histograms, and the data corresponding to PI, PE, and PS are shown in Appendix A. In these figures, PL species are denominated by their initials followed by the total number of carbons and the number of carbon–carbon double bonds in their acyl chains (as an example, PC 38:4 corresponds to a PC bearing 2 FA with a total of 38 carbon atoms and four double bonds).

As already described in previous studies [8], it was observed that PC is the PL species that displays the widest variations in terms of fatty acid chain distribution depending on the organ considered (Figure 1 and Appendix A), whereas PI is essentially represented by the species PI 38:4 in all the organs studied (Appendix A).

As previously reported for rats [6], some organs are enriched in PC species with PUFA acyl chains (e.g., PC 38:4), whereas other organs preferentially contain PC with two saturated fatty acyl chains (e.g., PC 32:0). These variations can be visualized by the DB index calculated for each organ (Figure 2), especially when PCA is applied to this index (Figure 3). As shown, skeletal muscle and cardiac tissue are particularly enriched in PUFA-containing PC species (DB > 2). By contrast, the brain and the lung contain remarkably high amounts of saturated PC species (DB = 0). The brain also differentiates from other organs by its high levels of monounsaturated PC species (DB = 1). In the latter case, PC 34:1 appears as the major species. The pancreas and, to a lower extent, the liver, also display a very characteristic signature, with high levels of DB = 2 PC.

Among PUFAs, important variations could also be observed depending on the organ considered (Figure 1 and Figure 2). If AA appeared as the most represented fatty acid in the brain, lung, and pancreas (see histograms in Figure 2), as combinations with palmitate (PC 36:4) or stearate (PC 38:4; Figure 1), DHA was exquisitely enriched in skeletal muscles and the heart, in combination with palmitate (PC 38:6) or stearate (PC 40:6; Figure 1).

Contrasting with PC, PE species containing PUFA were systematically dominant, whatever the organ considered (Appendix A). However, the DHA to AA balance greatly varied among organs. As already observed for PC, DHA was particularly enriched in skeletal muscles and in the heart, where PE 40:6 was the most represented species. Interestingly, DHA was also the most prominent PUFA in the brain, a situation very different from PC behavior in this organ (Appendix A). The lung and the pancreas appeared as the most DB = 2 enriched organs.

Concerning PS, the PS 40:6 species appeared as the most represented one in the brain and skeletal muscles (Appendix A). Due to little quantities of this PL class, PS fatty acid distributions could not be determined in the other organs with satisfactory accuracy and are therefore not displayed in the present study.

To summarize, all the organs studied here displayed very characteristic fatty acid distributions within PLs and various unsaturation rates: if one considers PC, they can be classified as DB = 0 (lung), DB = 1 (Brain), DB = 2 (Pancreas, liver) and DB > 2 organs (muscle and heart). When selectively focusing on PUFA, one can differentiate AA- (pancreas, lung, and brain) and DHA-enriched (heart and skeletal muscle) organs. The brain is the organ that displays the most discrepancies between PLs, PC being predominantly DB = 1, whereas PE and PS are essentially DHA-enriched. These observations match previous observations made in rats [6].

### 3.2. Recapitulating the In Vivo Lipid Signature In Vitro

Muscle cells undergo repeated cycles of elongation and relaxation and are thus constantly exposed to important mechanical constraints due to intrinsic repetitive muscle deformations [28] and are prone to experience increases in membrane tension [29]. Changes in membrane tension are known to be transduced into changes in surface area and cell volume [30]. If regulation of cell volume is a fundamental property of all animal cells, it is of particular importance in skeletal muscle, where exercise is associated with a wide range of cellular changes that influence cell volume [31]. Since the fatty acid composition of PLs is known to modulate the cellular membrane properties, and specifically their ability to deform under stretching/contraction [2,13], we, therefore, focused on the impacts of PL fatty acid composition on muscular cell adaptation to mechanical constraints.

With this aim, experiments of fatty acid supplementation have been performed on a simple in vitro muscle model, the C2C12 mouse cell line, known as a suitable model for studying muscle cells physiology [32]. First, C2C12 myoblasts were cultivated under usual conditions (no fatty acid supplementation; non-treated: NT), and the fatty acyl distribution within PC was determined. The results are displayed as radar graphs in Figure 4 and as histograms in Appendix A, and the corresponding DB indexes are presented in Appendix A. As shown, very low levels of PUFA-containing PC were observed under these conditions, in clear contrast to what is observed in muscles (Figure 1). Rather, PC under these conditions mainly corresponded to DB = 1 (PC 32:1 and PC 34:1) and DB = 2 species (PC 34:2 and PC 36:2). In order to modulate the cell composition in PUFA-containing PLs, C2C12 myoblasts were then cultivated in the presence of exogenous sources of PUFA. By contrast to saturated fatty acids (SFAs) and monounsaturated fatty acids (MUFAs), the body cannot synthesize PUFAs from scratch and must therefore obtain basic blocks from the diet, which are referred to as essential fatty acids (EFAs). As already mentioned, there are two families of EFAs, which are classified on the basis of the position of the first double bond from the last carbon of the acyl chain (the omega (ω) carbon): ω-3 and ω-6. The ω-6 building block is linoleic acid (18:2ω-6) and the ω-3’s is α-linolenic acid (18:3ω-3). These precursors are metabolized by a series of alternating desaturation and elongation reactions to form the main end-products AA (ω-6) and DHA (ω-3) [33]. End-PUFA products can also be directly obtained from the diet. For example, DHA is present in fish oils. Therefore, C2C12 myoblasts were cultivated in the presence of 100 µM of either the end-products AA and DHA or their precursors 18:2ω-6 and 18:3ω-3, and the distribution of these fatty acids within PC species was evaluated (Figure 4). As shown, AA addition resulted in the formation of the AA-containing PC species PC 36:4 and PC 38:4, showing that this FA was efficiently incorporated within PLs. Similarly, incubation with DHA induced the production of the DHA-containing species PC 38:6, which is the most prominent species in muscles (Figure 1). Interestingly, incubation with 18:2ω-6 and 18:3ω-3 did not lead to the optimal formation of the corresponding AA- and DHA-containing PC species, suggesting that these precursors were not efficiently subjected to the elongation/desaturation steps required to form the corresponding end-products in C2C12 myoblasts, at least efficiently enough to reproduce the lipid signatures observed in vivo (Figure 1). The same observation could be made for other PLs species PE, PS, and PI (data not shown). Therefore, in the following, we focused on three main culture conditions, i.e., C2C12 myoblasts cultivated either in the presence of 100 µM AA or DHA or under no fatty acid supplementation (NT).

### 3.3. PUFA Induce the Formation of Membrane Invaginations under Mechanical Constraints

To evaluate if cell membranes may respond differently to changes in area and shape depending on the fatty acyl composition of their PLs, we labeled C2C12 myoblasts with the FM1-43 probe, which becomes intensely fluorescent at 515 nm once inserted in the outer leaflet of the plasma membrane [16,17]. We observed membrane dynamics by modifying cell volume in response to exposure to a medium with a 55% reduction in osmolarity (osmotic downshock). Under similar conditions, fibroblasts or neurons have been shown to form dome-shaped micrometer-sized invaginations termed vacuole-like dilations (VLDs), which have been hypothesized to constitute a mechanism to accommodate excess membrane area upon osmotic-induced cell shrinking [34,35]. It has been hypothesized that, during osmotic downshock, the flowing bilayer of nascent VLDs may be water permeable and increasing intracellular water could exit the cell and fill VLDs at a high rate [36]. Since VLD formation has been shown to be fully passive [34], we hypothesized that membrane biophysical properties, which are highly dependent on the fatty acid composition of their PLs, could directly influence the formation of such structures.

Accordingly, a first important observation was that C2C12 did not form VLDs with the same efficiency depending on the culture condition considered. Some representative images corresponding to FM1-43-labeled myoblasts cultivated either in DHA-supplemented or not-supplemented (NT) media and submitted to osmotic shock are displayed in Figure 5A. As shown, whereas VLDs could be clearly visualized under DHA supplementation, as plasma membrane invagination emerging from the basal substrate-bound side of the membrane (Figure 5B: 3D view), most NT myoblasts failed to form such structures. Interestingly, quantification of the average number of VLDs per cell revealed that the nature of the PUFA added to the medium also greatly influenced the number of these membrane invaginations: if the ω-3 PUFA DHA resulted in the formation of 11.4 ± 0.7 (*n* = 148) VLDs per cell, the ω-6 AA resulted in an average number of 4.2 ± 0.4 (*n* = 103) VLDs per cell, a number very similar to the one observed under NT conditions (3.8 ± 0.3 (*n* = 152); Figure 5C). Some differences between NT and AA-myoblasts could, however, be visualized when considering the percent distribution of cells as a function of their VLDs-number (Figure 5D): AA-myoblasts differentiated from NT by forming preferentially between 1–5 VLDs per cell (48%), 34% of NT myoblasts forming no VLD at all. By contrast, half of the DHA-myoblast population formed more than 10 VLDs per cell (Figure 5D). These results suggest that the cellular PL composition influences the ability of the cells to form VLDs under osmotic stress.

In the next step, we evaluated the kinetics of VLD formation under various conditions (Figure 5E–H and Appendix A). VLD formation was measured from mean fluorescence counted in a circular region of interest surrounding the VLD (Figure 5E). Fluorescence kinetics obtained reflects VLD formation kinetics (Figure 5F). From these measurements, we extracted two parameters describing the overall kinetics of VLDs formation under the various conditions, i.e., the t_Vmax_, which corresponds to the time when V_max_ (the maximal velocity of VLDs formation) is reached (Figure 5F). As shown in Figure 5G, incubation with PUFA, either AA or DHA, resulted in a reduction of t_Vmax_ as compared to NT myoblasts (NT: 622 ± 21 s; AA: 423 ± 25 s; DHA: 301 ± 10 s). By contrast, only DHA resulted in a significant increase in V_max_ (Figure 5G; NT: 0.016 ± 0.001 s^−1^; AA: 0.018 ± 0.002 s^−1^; DHA: 0.024 ± 0.002 s^−1^). Therefore, if both PUFA, DHA, and AA, trigger an early onset of VLDs formation in response to osmotic downshock as compared to non-supplemented conditions, DHA displays the specific property of accelerating the process.

### 3.4. Forming Membrane Invaginations Reduces Membrane Breakage under Mechanical Constraints

Another interesting characteristic of the FM 1-43 dye is that it can enter the cell through open mechanotransduction channels [18] or when more drastic conditions are reached, i.e., under plasma membrane rupture and cell breakage (Appendix A). Under such conditions, FM1-43 rapidly fills the cytoplasm and generates an intense fluorescence at 543 nm (Figure 6A, Appendix A, and [18]). Therefore, we used this property to evaluate plasma membrane integrity under various conditions. A first observation was that myoblasts forming VLDs were less susceptible to FM1-43 labeling of their cytoplasm as compared to cells deprived of such structures (as an example, see Figure 6B). In order to obtain a quantitative view of this process, using the same osmotic downshock, we recorded fluorescence images at 543 nm, and mean fluorescence was measured in each cell. As shown in Figure 6C, PUFA-treated myoblasts were less susceptible to perturbations in membrane integrity under osmotic shock than non-treated cells. This resistance process clearly correlated with myoblast ability to form VLDs, as monitored by the maximal relative fluorescence obtained in labeled cells as a function of the number of VLDs (Figure 6D). Indeed, the maximal relative fluorescence (Fmax-F0/F0) values were obtained in cells unable to produce VLDs with relative fluorescence values 3-fold higher as compared to cells that were able to produce VLDs. It seems that a few numbers of VLDs (between 1 to 5 VLDs per cell) are sufficient to maintain low fluorescence values.

We next questioned whether differential susceptibility to VLDs formation or cell breakage could be correlated to different cell surface variations under the various conditions tested in response to osmotic shock (Figure 6E). In this figure, we differentiated among myoblasts cultured under non-supplemented conditions the ones which have resisted cell breakage by forming VLDs (NT VLD) from the ones displaying FM1-43 yellow labeling of their cytoplasm and no VLDs (NT cracked). As shown, the variations in cell surface were comprised between 12.6 ± 2.8% (*n* = 96) for AA and 19.5 ± 4.0 for NT cracked (*n* = 58), with no significant variations among the different conditions (Figure 6E). Interestingly, however, when considering the time following exposure to osmotic shock leading to maximum surface variation (t_Surface max_), DHA treatment appeared to accelerate the obtention of this maximum (Figure 6F): this maximum was reached after 137.7 ± 6.6 s with DHA (*n* = 88), whereas 227.3 ± 14.0 s were required to reach such maximum in NT, VLD-forming C2C12 (*n* = 47). Altogether, these data show that the various culture conditions do not seem to influence the maximal cell surface reached under osmotic shock, which corresponds to an approximately 20% increase as compared to the initial surface, whatever the condition considered. However, treatment with DHA appears to accelerate the achievement of this maximal surface in response to cell swelling.

Finally, to obtain a global overview of the overall response to osmotic shock, the t_0_ of VLDs formation (t_0_ VLD; Figure 5F or Figure 6G) and the time of breakage of NT myoblasts (t breakage; Figure 6H) were also determined. The sequence of events based on the data displayed in Figure 5 and Figure 6 is recapitulated in Figure 7A.

As shown, if both PUFA AA and DHA proved to be efficient in reducing cell breakage under osmotic shock (Figure 6C), they clearly promoted different cell behaviors in response to cell swelling. DHA accelerated the obtention of the maximal cell surface (t_Surface max_), the time of initiation (t0 VLD), and the rate of VLDs formation (V_max_) and, finally, increased the overall number of VLDs per cell. AA displayed an intermediate behavior between the NT and DHA conditions. Interestingly, in the NT myoblast unable to form VLDs, cell breakage (Figure 6H) was observed at a time following osmotic shock posterior to the time when AA and DHA myoblasts started to form these structures (Figure 6G; recapitulated in Figure 7A). These data are schematized in Figure 7B.

### 3.5. Adaptation of PUFA-Containing PLs to Surface Expansion

To draw a possible mechanism accounting for the different behaviors of NT-, AA-, and DHA-treated cells in response to osmotic shock, the behavior of membranes composed of AA- and DHA-containing PLs was evaluated using molecular dynamics simulations under stretching. Coarse-grained simulations were performed using the Martini forcefield [21]. In all simulations, we varied the acyl chain composition while keeping the PC polar head constant. We used five lipids to assess the effect of PUFA on biophysical properties of the membrane under extension and their mixes: the DB = 0 species DPPC (1,2-dipalmitoyl-sn-glycero-3-phosphocholine; (PC 32:0) or PC (16:0-16:0)), the DB = 1 POPC (1-palmitoyl-2-oleoyl-sn-glycero-3-phosphocholine; PC 34:1; PC (16:0-18:1)), the DB = 2 DOPC (1,2-dioleoyl-sn-glycero-3-phosphocholine PC 36:2; PC (18:1-18:1), and the DB > 2 species PAPC (1-palmitoyl-2-aracidonoyl-sn-glycero-3-phosphocholine; PC36:4; PC (16:0-20:4)) and PUPC (1-palmitoyl-2-docosahexaenoyl-sn-glycero-3-phosphocholine; PC 38:6; PC (16:0-22:6)). These corresponded to the main species encountered in the various organs (Figure 1) or in myoblasts grown under the various fatty acid supplements (Figure 4). Since POPC membranes systematically displayed an intermediate behavior between DPPC and DOPC bilayers in all the simulations performed in this study, we chose not to show the corresponding data to simplify the reading. To mimic a membrane extension, several simulations were performed by constraining the X and Y value of the box (membrane plane) to remain constant during the simulation (Figure 8A). Using this method, we were able to perform membrane extension from 5% up to 25%, a range that covered the percentage increase of myoblast surface under osmotic shock (Figure 6E). The first parameter that we evaluated in this model was membrane tension [37,38]. Indeed, physiological conditions, such as cell swelling, which is associated with an increase in membrane surface, resulting in membrane tension, a parameter that in turn regulates membrane dynamics [37,38]. As shown in Figure 8B, membrane extension clearly correlated with an increase in membrane tension in this model. However, even if the presence of PUFA in PC species seemed to slightly decrease surface tension under surface expansion as compared to pure DOPC bilayers, no significant variations were observed, suggesting that the presence of PUFA, either AA (PAPC) or DHA (PUPC), are not strong reducers of membrane tension per se. The exact same behaviors were observed when DOPC was replaced by either DPPC or POPC (data not shown). To release membrane tension as cells change shape, the plasma membrane has to adapt to the new shape and the membrane area must be redistributed accordingly. This redistribution comes primarily from the readily available membrane reservoir, or buffer, present at the surface of the cell in the form of membrane superstructures, including VLDs [37,38]. Since PUFA within PLs can act as contortionists to adjust membrane deformation [12], we evaluated their behavior in response to surface extension. To do so, the angle of the acyl chain was computed as the angle between two vectors (Figure 8C). The first vector is defined by the first and last beads of the acyl chain. The second vector is the perpendicular axis to the membrane plane. Therefore, angles close to 0° mean that the considered acyl chain of the PL is perpendicular to the membrane plane. By contrast, when the angle is close to 90°, the acyl chain is positioned in parallel to the membrane plane. The probability of reaching a given angle (angle α) for the considered fatty acyl chain was then represented as a function of the extension rate. Representative experiments are presented in Figure 8D–F. As shown in Figure 8D, when considering a saturated fatty acyl chain such as palmitate, at the sn-1 position of PC, the distribution of the angle α to be quite restricted to values around 30° (or 180 − 30 = 150° for same PC species distributed in the opposite leaflet of the membrane bilayer; Figure 8C) at equilibrium (i.e., when no extension was applied). This distribution was very similar whether another palmitate chain (DPPC) or a PUFA such as DHA (PUPC) occupied the sn-2 position. Upon extension, a redistribution to angles of higher values could be observed, with a probability of approximately 0.02 being reached for α angles of 90°, showing that a proportion of the palmitoyl chains of PC can switch from almost perpendicular to the membrane plane, to horizontal relative to this latest. Interestingly, upon extension, the fact that a DHA, rather than palmitate, occupied the sn-2 position appeared to favor the horizontal positioning of the sn-1 palmitoyl chain (Figure 8D). AA displayed a different behavior compared to palmitate (AA at the sn-2 position, in combination with palmitate; PAPC), in the sense that its angle distribution was more even at equilibrium than the one of palmitate (Figure 8E) to reach very similar probabilities in the range of 50–90° upon maximal extension (Probability 0.035–0.040). Interestingly, this behavior appeared to be quite independent of the fatty acyl chain composition of the neighboring PLs since very similar results could be obtained with DPPC/PAPC or DOPC/PAPC mixes (Figure 8E). Therefore, the ability to extend in the plane of the bilayer under extension appeared to be an intrinsic property of the AA chain. An even more extreme situation could be observed with DHA (Figure 8F). Indeed, high probabilities in the 50–90° area were observed under equilibrium, reaching a value of 0.06 under maximal extension (Figure 8F) to correspond to the main conformation under these conditions. As observed for AA, DHA appeared to be quite insensitive to the nature of the fatty acyl chain of the surrounding PLs. To summarize this part, PUFA, by contrast to saturated chains, displays the property of switching their conformation from rather perpendicular to the bilayer plane at equilibrium to parallel to this latest upon extension. The ω-3 PUFA DHA appears to be more prone to occupy the parallel conformation as compared to the ω-6 PUFA AA, a behavior that is even amplified under membrane extension.

If DHA tends to preferentially position in parallel to the plane of the bilayer under membrane expansion, we next questioned the level of extension of the chain under these conditions. With this aim, the distance between the first and the last beads was calculated in each case (dist; Figure 8G). In the region of 80° < Angle α < 100°, the DHA was mainly found under two favorite conformations, a compact one, characterized by 0.5 Å < dist < 0.6 Å, and an expanded one (0.9 Å < dist < 1.0 Å). These data show that DHA not only displays a high propension to position in the plane of the bilayer upon expansion (Figure 8F) but also to expand its chain in this plane. As such, acting as a contortionist, it offers to the membrane the ability to expand in the surface more easily than saturated fatty acid chains.

## 4. Discussion

In the present study, we first confirmed previous studies showing that mouse organs display very specific fatty acid signatures within their PLs [7]. The skeletal muscle and the heart differentiate from other organs since they display a selective enrichment of the ω-3 PUFA DHA within their PC species (Figure 1, Figure 2 and Figure 3), raising the question of the physiological relevance of this signature.

The heart and the muscle have in common that they are subjected to important variations in size and volume when accomplishing their physiological function [28]. The control of cell size is complex and involves full protein machinery [39]. However, the first responses to mechanical stress appear to be largely independent of this machinery and are driven by passive processes regulated by the mechanical properties of the cell per se [34]. Knowing that the fatty acyl composition of PLs highly contributes to the definition of cell membrane properties, in particular in terms of their ability to deform [2], one may therefore speculate that this specific signature is crucial to sustaining selective functions in these organs. Accordingly, the skeletal muscle has been shown to be highly susceptible to fatty acid remodeling of its PC in response to diet, in a process that correlates with alterations in its contractile function [6].

In an attempt to evaluate the effects of PUFA-containing PLs on muscular cell adaptation to mechanical constraints, we, therefore, in a first step, tried to recapitulate at best the PL signature observed in vivo in cultured myoblasts. A first important observation was that myoblasts cultured under usual conditions are fully deprived of PUFA-containing PL species. This was not a surprising observation since PUFAs have to be produced from precursors, known as essential fatty acids (EFAs), which can only be made available from exogenous sources (i.e., the diet). It has already been shown that the amounts of EFAs present in the serum, which is generally added to the culture medium, are not sufficient to reconstitute the PUFA levels in PLs observed in vivo, even if some sera may prove to be more efficient than others [40,41]. Moreover, as shown in the present study, precursors, even if provided in important amounts, may not be correctly metabolized to the corresponding end-products (Figure 4 and Figure 5) since their elongation and desaturation require additional steps which may not be turned on, depending on the kind of cell used. Such a low conversion efficiency of EFA precursors to their final products has been observed in vivo [42]. For example, the conversion of ingested 18:3ω-3 to DHA is very inefficient in Man, with <10% conversion in women and <3% in men [43,44]. This is likely due to the fact that a specific desaturase, namely ∆6 desaturase, is rate-limiting [45]. Considering the above, providing the cells with end-product PUFAs in the medium appears as a suitable solution to circumvent these bottlenecks and mimic the PL signature encountered in vivo (Figure 4 and Figure 5).

In good agreement with the formulated hypothesis, C2C12 muscle cells displaying various FA-containing PLs showed very different behaviors when confronted with mechanical constraints, under the form of an osmotic downshock (Figure 5). In response to this stress, C2C12 cells containing PUFA-PLs formed tube-shaped invaginations known as vacuole-like dilations (VLDs; [34,35]). Since VLD formation has been shown to be fully passive [34], these data suggest that the fatty acyl composition of membrane PLs largely influences the ability of the cell to form such structures under mechanical constraints. Interestingly, VLD formation also appeared to be protective for the cell since the formation of such structures reduced membrane breakage under osmotic downshock (Figure 6). Altogether, these observations suggest that PUFA-containing lipids are important actors of cell adaptation to mechanical constraints by avoiding their shrinkage via facilitated VLD formation (Figure 7).

Cell swelling or stretching result in a transient increase of the plasma membrane surface. In this study, we evaluated the surface variation of C2C12 cells to be comprised between 12% and 20% depending on the fatty acyl composition of PLs under osmotic downshock (Figure 6E). Molecular dynamics simulations revealed that such a surface increase results in a dramatic augmentation in surface tension (Figure 8B). Surface tension is related to the force needed to deform a membrane [46]. In cells, the force needed to deform the plasma membrane is greater than that for a pure lipid bilayer, as modelized here, due to contributions from membrane proteins and membrane-to-cortex attachments, which link the membrane to the underlying cortex and also resist membrane deformation [46]. In short, under osmotic downshock, the plasma membrane is subjected to huge mechanical constraints that should be released to avoid breakage.

One may postulate two main strategies to reduce surface tension, which is, in a simplified way, the result of stretching constraints on individual membrane components, among which are PLs: an inflow of PLs to the membrane, either by neosynthesis (which is not likely to occur under such a very short period of time) or by the mobilization of membrane reservoirs, and/or an intrinsic ability of membrane constituents to deform to adjust and soften these constraints.

At the difference of SFAs or MUFAs, PUFAs display the ability to contortionate in the lipid bilayer, therefore softening various mechanical stresses in the membrane [12,13]. This is due to the chemical feature of PUFA, which displays saturated carbons (CH_2_) surrounded by two unsaturated ones (=CH-CH_2_-CH=). As a consequence, rotational freedom around these CH_2_ groups is extremely high as compared to rotations around the CH_2_ groups of SFAs and MUFAs [12].

Interestingly, at least based on the molecular dynamics simulations performed in this study, the presence of PUFA within PLs tends to slightly reduce surface tension under membrane extension (Figure 8B). PUFA-containing PLs display the ability to switch their position from rather perpendicular to parallel to the membrane plane under surface expansion (Figure 8E,F). This behavior clearly differentiates from the one observed with the SFA palmitate and is more pronounced for DHA than for the ω-6 PUFA AA (Figure 8D–F). Moreover, DHA not only displays a high propension to position in the plane of the bilayer upon expansion (Figure 8F) but also to expand its chain in this plane (Figure 8G). This ability to deform makes DHA a quite good candidate to soften the impacts of increased surface tension under osmotic downshock. Moreover, this combination of behaviors results in an increase in the area per lipid, as already observed by others [47], and therefore, for a membrane with a fixed number of PLs, in its overall surface expansion. Based on the data presented here (Figure 8G) and in previous studies [47], DHA, by switching from perpendicular to fully expanded in the plane of the bilayer, can induce a two-fold increase in DHA-PL area per lipid. Since DHA-PLs constitute approximately 25% of total PLs in DHA-C2C12 (Figure 1), they could account, by themselves, for a 25% membrane surface increase if switching collectively between these two extreme conformations under stretching, therefore compensating largely the surface increase observed under osmotic downshock, without the need of additional PL influx to the membrane.

Another important characteristic of membrane-containing PUFA-PLs is that they tend to form thinner and longer tubes than membranes composed of SFA- and MUFA-PLs when submitted to a punctual pulling force that mimics tension through membrane-to-cortex attachments [12]. This shows that a lower pulling force is required to form a tube from a PUFA-PL containing membrane. Again, among PUFAs, DHA appears to be more effective in this process than AA [12]. These observations nicely fit the data obtained in the present study, showing that VLDs tend to form much faster in DHA-PL-containing cells, as visualized by a lower t_Vmax_ VLD (Figure 5G) and a higher V_max_ VLD (Figure 5H), as compared to other conditions.

To summarize, due to their specific conformational properties, PUFA-PLs, and particularly DHA-PLs, are strong promoters of VLD formation in a process that is likely due to their ability to accompany surface expansion and to reduce the energy (pulling force) required for them to form. Since VLD formation clearly correlates with cell resistance to breakage under mechanical constraints, these data therefore suggest that DHA enrichment in muscle and heart may have been selected during evolution to match at best the mechanical constraints and variations in volumes these specific tissues are confronted to.

## Figures and Tables

**Figure 1 cells-10-00937-f001:**
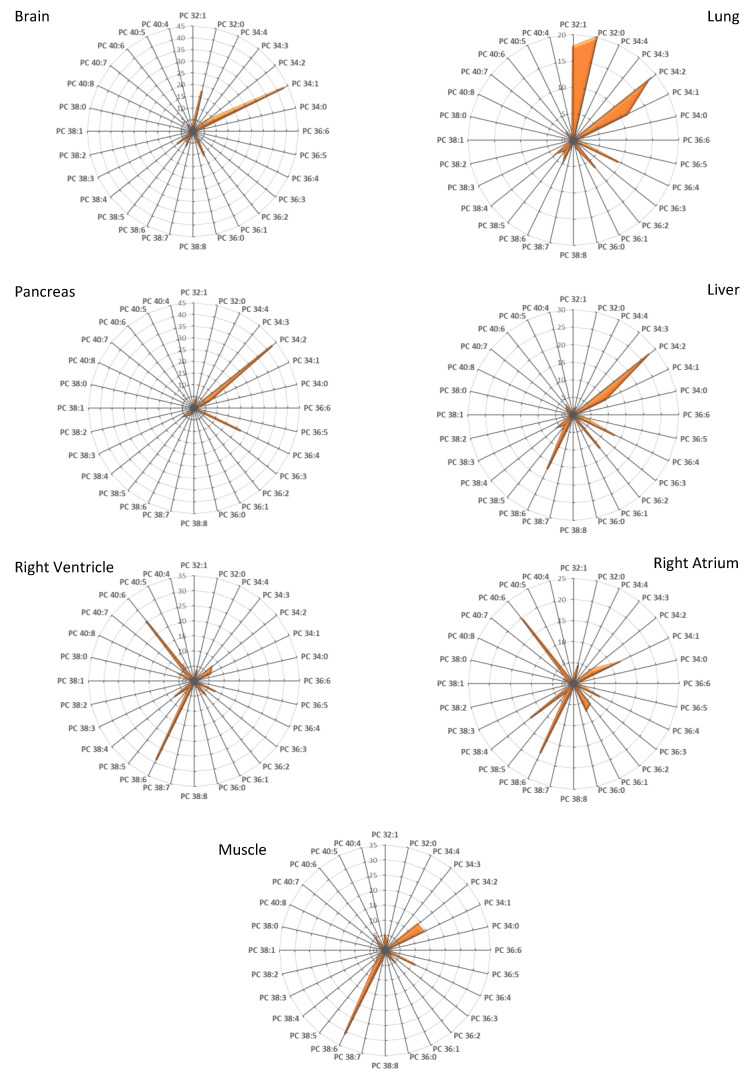
PC species distribution in various organs excised from mice. PC subspecies distribution in each case is displayed. The total carbon chain length (x) and number of carbon–carbon double bonds (y) in the acyl chains of the main PC molecular species (x:y) are indicated. Values are means of four independent determinations from four individuals from both groups in each case.

**Figure 2 cells-10-00937-f002:**
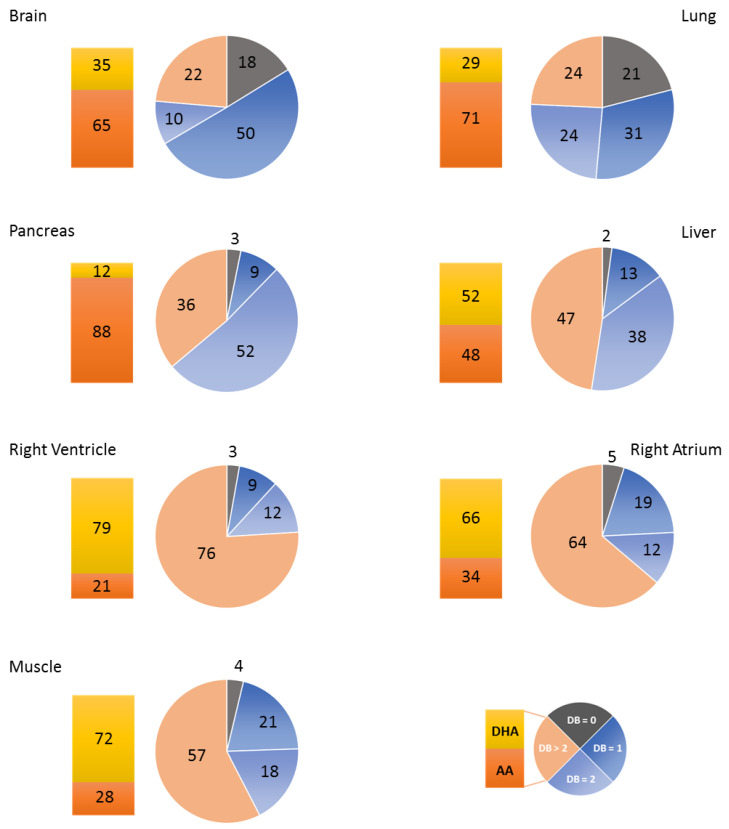
PC double bond (DB) index and docosahexaenoic acid (DHA) to arachidonic acid (AA) ratios in various organs. The relative percentage of PC species bearing no double bond (DB = 0), one double bond (DB = 1), two double bonds (DB = 2), and > two double bonds (DB > 2) in their acyl chains were obtained from the PC subspecies distribution displayed in Figure 1. The ratio of DHA (PC 36:6, 38:6, and 40:6) to AA (PC 36:4, 38:4, and 40:4)-containing PC subspecies in the various organs is also displayed and was calculated from the values presented in Figure 1. The percentages of the various subspecies are indicated.

**Figure 3 cells-10-00937-f003:**
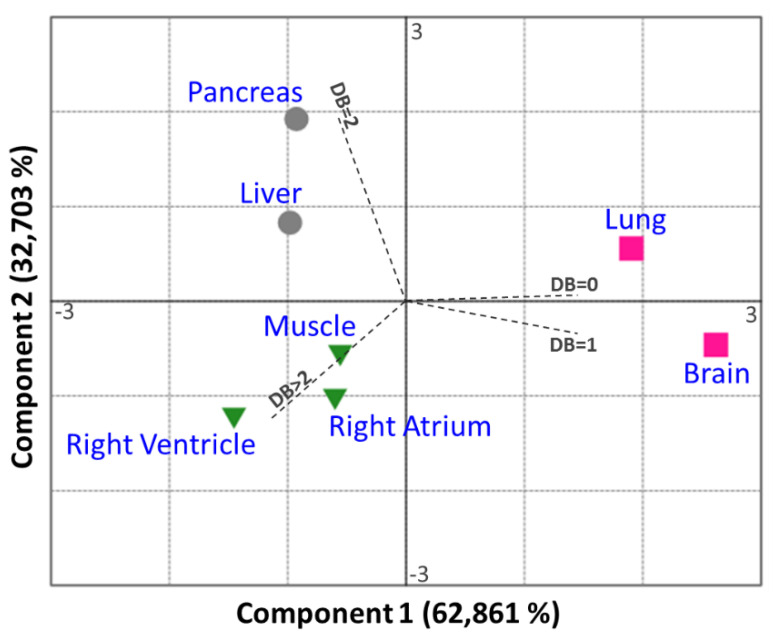
Principal component analyses (PCA) of the variability in PC DB index as a function of the organ. The variables measured in this study (i.e., DB = 0, DB = 1, DB = 2, and DB > 2) are expressed as vectors in dashed lines.

**Figure 4 cells-10-00937-f004:**
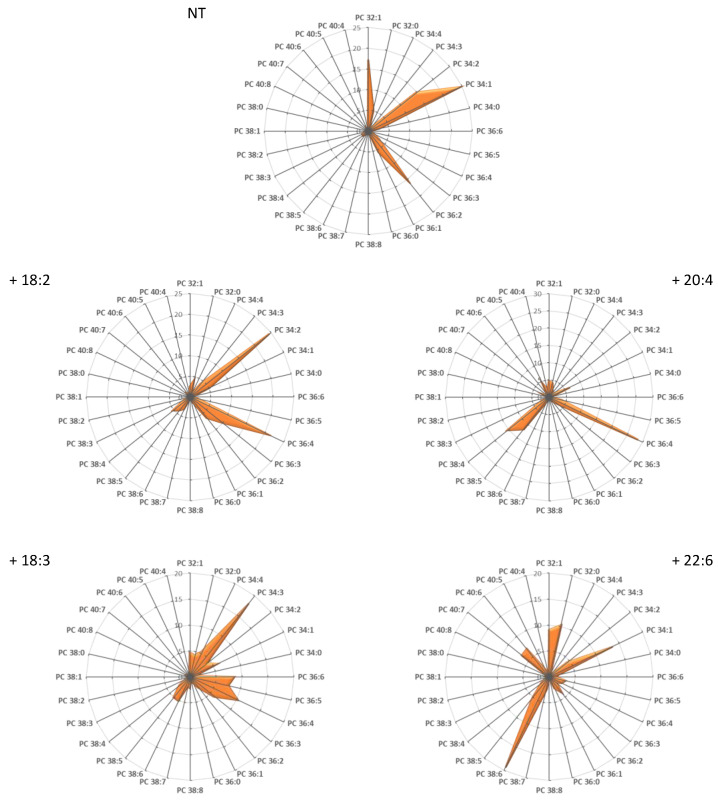
C2C12 incorporate differently exogenous fatty acids within their phospholipids. C2C12 myoblasts were cultivated for 16 h in the presence of 100 µM of the indicated fatty acids. PC subspecies distribution in each case is displayed. The total carbon chain length (x) and number of carbon–carbon double bonds (y) in the acyl chains of the main PC molecular species (x:y) are indicated. Values are means of four independent determinations in each case.

**Figure 5 cells-10-00937-f005:**
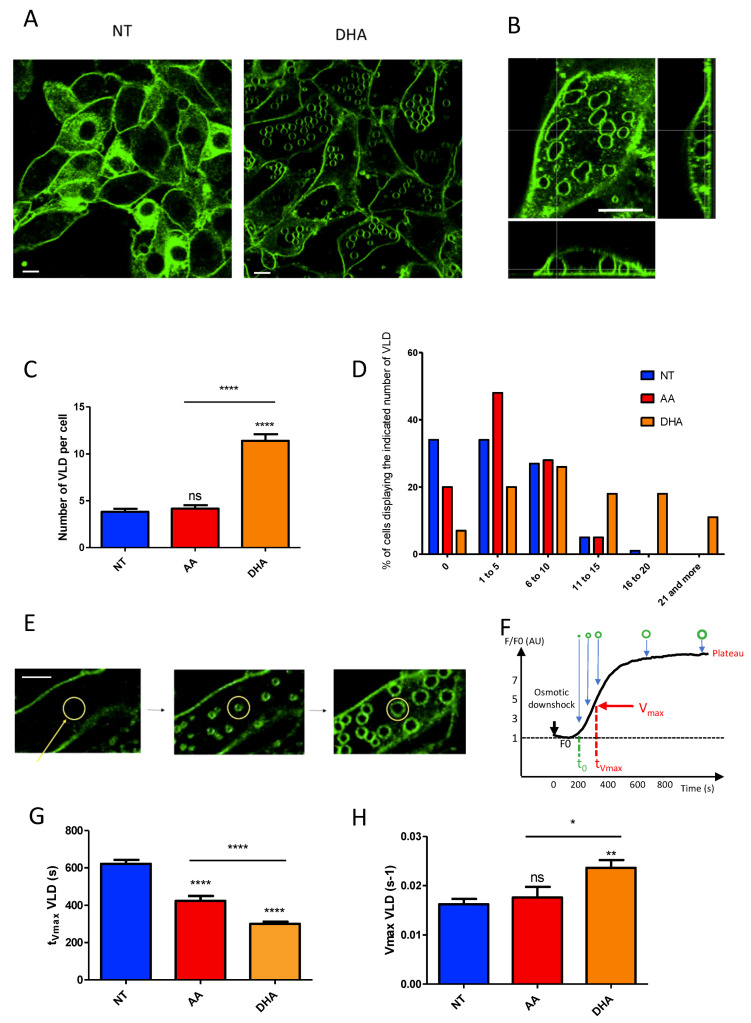
C2C12 respond differently to osmotic downshock depending on the nature of the fatty acid composition of their membrane phospholipids. (**A**) Representative fluorescence images (515 nm) of FM1-43-labeled myoblasts cultivated either in DHA-supplemented or not-supplemented (NT) media and submitted to osmotic shock. Scale bar: 10 µm. (**B**) 3D view of a cell exhibiting vacuole-like dilations (VLDs). Scale bar: 10 µm (**C**) Number of VLDs per cell in AA, DHA and not-supplemented cells. (**D**) Percent distribution of cells as a function of their VLDs-number in AA, DHA, and not-supplemented cells. (**E**) Examples of images showing VLDs formation before (left), during (middle), and at the end of formation (right). Scale bar: 5 µm. Mean fluorescence was counted in a circular region of interest (yellow circle) and reported as a function of time (**F**). Several parameters were calculated from these curves: amplitude (plateau) and kinetic parameters (t_0_: time to fluorescence appearance; V_max_: maximal rate of fluorescence increase; t_Vmax_: time to V_max_). (**G**,**H**) t_Vmax_ and V_max_, respectively, obtained in AA, DHA, and not-supplemented cells. Data are presented as mean ± SEM. Statistical analysis was performed using two-way ANOVA, completed by Bonferroni post-tests to compare means variations between the various conditions. Significant differences are indicated as **** *p* < 0.0001, *** *p* < 0.001, ** *p* < 0.01 and * *p* < 0.05.

**Figure 6 cells-10-00937-f006:**
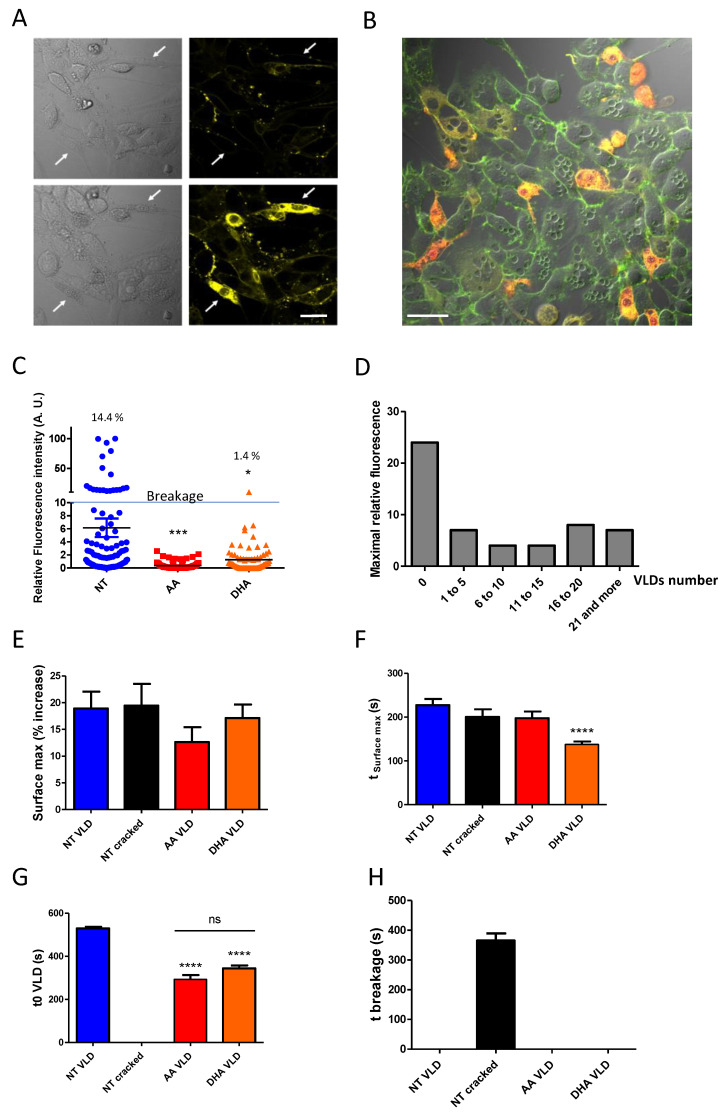
VLD formation limits cell breakage in a process that is modulated by the fatty acid composition of membrane phospholipids. (**A**) Examples of cell breakage following osmotic downshock: when the integrity of the cell membrane is lost, FM1-43 rapidly fills the cytoplasm and generates an intense fluorescence at 543 nm. Representative transmission (left) and fluorescence (543 nm) images (right) of FM1-43-labeled myoblasts before (above) and after (below) osmotic downshock. White arrows designate two cell breakages (for dynamic view, see Appendix A). Scale Bar: 20 µm. (**B**) Composite image of fluorescence images recorded at 515 and 543 nm showing both VLDs and cells with high yellow fluorescence. Scale Bar: 20 µm. (**C**) Relative fluorescence intensity at 543 nm obtained at the end of the osmotic downshock. The % displayed indicate the percentages of C2C12 exhibiting relative fluorescence intensities above 10 under the various culture conditions. This fluorescence value corresponds to bona fine cell breakage. (**D**) Maximal relative fluorescence obtained in labeled cells in all conditions as a function of the number of VLDs. (**E**,**F**) The maximal variations in cell surface (**E**) and the time following osmotic downshock when these maximal surfaces were reached (t_Surface max_; **F**) are indicated for the various conditions. (**G**) t0 of VLD formation (t0 VLD; t0 in Figure 5F). (**H**) t breakage, corresponding to the time when cell breakage occurred following the initiation of osmotic downshock. Data are presented as mean ± SEM. Statistical analysis was performed using two-way ANOVA, completed by Bonferroni post-tests to compare means variations between the various conditions. Significant differences are indicated (**** *p* < 0.0001, *** *p* < 0.001, ** *p* < 0.01 and * *p* < 0.05).

**Figure 7 cells-10-00937-f007:**
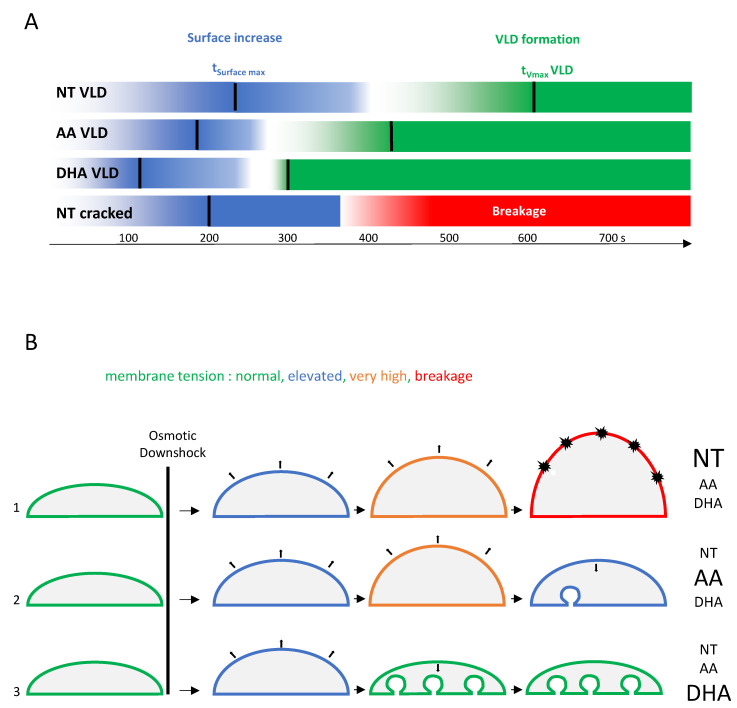
Summary of the sequence of events leading to cell resistance to breakage in response to osmotic downshock. (**A**) Schematic and recapitulative time sequences of surface increase (blue) and VLD formation (green) in NT C2C12 displaying VLDs, NT cracked C2C12 without VLDs, and C2C12 cultivated in media supplemented with either AA or DHA. Time sequence of breakage is also reported (red) for the NT cracked cells. Black bars are positioned at the time values for t_surface max_ and t_Vmax_ for VLDs and are extracted from data in Figure 5and Figure 6. (**B**) C2C12 cells may experience three different scenarios following osmotic downshock depending on their membrane composition. 1: membrane tension increases progressively until breakage (dark explosions), frequently observed in NT cells. 2: When membrane tension reaches a certain threshold (very high), some VLDs form at the basal membrane level, allowing the reduction of membrane tension, frequently observed in AA-treated cells. 3: In a scenario particularly encountered in DHA-treated cells, many VLDs are forming, leading to very low membrane tension, a process that avoids cell breakage.

**Figure 8 cells-10-00937-f008:**
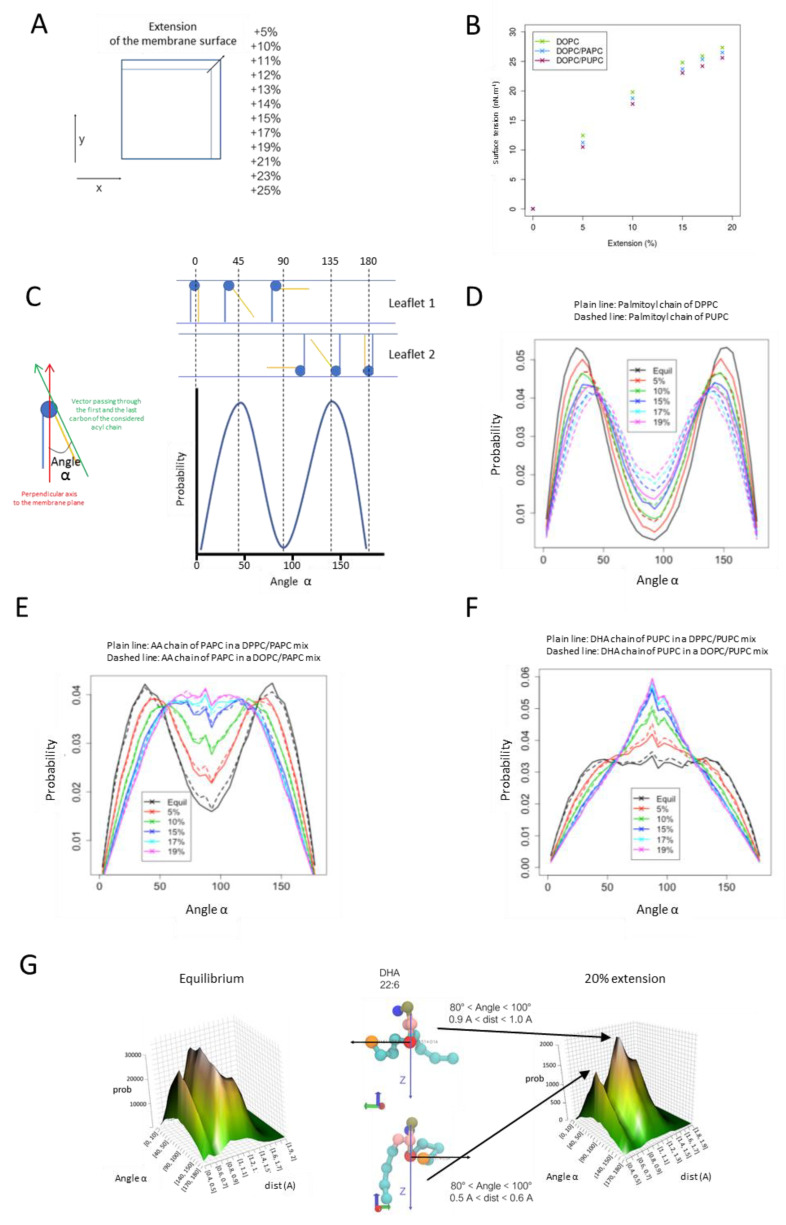
Molecular dynamics simulations of simple membranes composed of PC bearing various fatty acid compositions in response to surface expansion. (**A**) To mimic surface expansion using molecular dynamics simulations, the values of the dimensions x/y of the system, corresponding to the membrane plane, were artificially increased. A first simulation is conducted at equilibrium, then the x/y is increased by 5% and blocked, so the membrane surface remains the same during the simulation (pressure is kept constant at 1 bar with the dimension z of the system). (**B**) Evolution of the surface tension as a function of surface expansion for membranes composed of PC species bearing various fatty acid compositions. DOPC: PC bearing two oleoyl chains; PAPC: AA at the sn-2 position of PC, in combination with palmitate; PUPC: DHA at the sn-2 position of PC, in combination with palmitate. (**C**) The angle α represents the angle between two vectors. The first vector is defined by the first and last beads of the acyl chain. The second vector is the perpendicular axis to the membrane plane. Angles close to 0° mean that the considered acyl chain of the PL is perpendicular to the membrane plane. By contrast, when the angle is close to 90°, the acyl chain is positioned in parallel to the membrane plane. Due to the two leaflets of a membrane, there are two symmetrical distributions. (**D**) Probabilities of the angle α to reach a given value for a palmitoyl chain as a function of surface expansion. Plain lines show the distributions of angles for a palmitoyl chain in DPPC (sn-1 position). Dashed lines show the distributions of angles for the palmitoyl chain of PUPC. DPPC: PC bearing two palmitoyl chains. (**E**) Probabilities of the angle α to reach a given value for an AA chain as a function of surface expansion. Plain and dashed lines show the distributions of angles for the AA chain from PAPC when mixed with either DPPC or DOPC, respectively. (**F**) Probabilities of the angle α to reach a given value for a DHA fatty acyl chain as a function of expansion surface. Plain lines show the distributions of angles for the DHA chain from PUPC lipids mixed with DPPC. Dashed lines show the distributions of angles for the DHA chain from PUPC lipids mixed with DOPC. (**G**) 3D probability graph of the DHA chain conformation in PUPC, either at equilibrium (left) or under 20% extension (right), as a function of the angle α and the distance between the first and last beads of the DHA chain (DOPC/PUPC mix). Representative conformations of the DHA chain of PUPC are displayed.

## Data Availability

The datasets generated during and/or analyzed during the current study are available from the corresponding author on reasonable request.

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
