# Peer review of "Polyunsaturated Phospholipids Increase Cell Resilience to Mechanical Constraints"

_cells, 2021, doi:10.3390/cells10040937_

Round 1
Reviewer 1 Report
Kadri et al. revealed the mechanical properties of polyunsaturated phospholipids-containing membranes. To achieve this aim, rst, the authors investigated lipid (especially PC) species distribution in various organs from mouse. In particular, they focus on the number of double bonds in lipid (DB index), DHA/AA ratio, and DB index as a function of the organ. Next,
they also investigated lipid species distribution in C2C12 myoblast incorporated differently exogenous fatty acids. Base on the lipid species distribution, they revealed C2C12 response to osmotic downshock. For C2C12 cultivated in DHA-supplemented media, VLDs formation was observed and they carefully analyzed the VLDs formation kinetics and cell breakage. Finally, to explain the mechanical properties of membranes, they performed MD simulations.
The amount of experiments and simulations is enormous in this manuscript, and I have the impression that the research is being carried out carefully. However, there are so many questions and de ciencies in the manuscript. Therefore, I cannot recommend publishing this version in Cells. My comments are listed below and I hope the following comments will help the improvement of the manuscript.
1. I recommend that PolyUnsaturated Fatty Acid should be abbreviated as PUFA and PolyUnsaturated Fatty Acids should be abbreviated as PUFAs. The other abbreviations also should be revised as a same manner.
2. At Line 20, Faty to Fatty.
3. At Line 23, the authors should add some references.
4. At Line 34, the authors should explain about Ref.[5] concretely, not just cited.
5. From Line 36 to 55, the authors explain based on only Ref.[5]. Since Ref.[5] is the paper by the author's group, the authors should introduce the research background and motivation with comparing the other group studies.
6. At Line 60, the research motivation that PUFAs are included in muscle cells is not sufficient. Explain about the research motivation more carefully. I cannot understand why the authors think \DHA-containing PLs could be central players for muscle cell adaptation to mechanical constraints.".
7. At Line 90, 3 of mm3 should be superscript. There are unusually many places where subscripts and superscripts are not reflected, making it difficult to read the manuscript. I do not point out such mistakes in this report.
8. At Line 102, delete a â—¦C.
9. At Line 104, add the numbers of carbon and double bond for AA and DHA, for example, Arachidonic Acid (AA, 20:4).
10. At Line 113, ml to mL.
11. At Line 199, 15 is a relative permittivity? If so, why was this value used. In general,
this value is too small for water and too high for hydrocarbon region.
12. At Line 202, systems to system and were to was.
13. At Line 208, performed to perform.
14. At Line 212, what is g?
15. Since the authors discussed the PC species distribution based on the number of double bond, I suggest that the order of lipids is sorted by the number of double bond in Fig.1. Otherwise, display using different colors depending on the number of double bond.
16. In Fig.2, values should be shown in circle graphs and bar graphs.
17. In Fig.2, how did the authors obtain the AA/DHA ratio?
18. At Line 269, Fig.2 ! Fig.1.
19. In Fig.3, the values in B and C are too small to read.
20. At Line 306, the authors wrote \If Arachidonic Acid (AA; 20:4) appeared as the most represented fatty acid...". What is the basis for thinking this way?
21. At Line 311, why did not the authors mention about PI here?
22. At Line 319, the sentence of \PS fatty acid distributions could not ...." should appear also in the caption of Supplementary Figure 4.
23. At Line 338, what is the important mechanical constraints. Explain more concretely.
24. At Line 340, whether does this \volume" mean membrane volume or cell volume?
25. In Supplementary Figure 5, \None" ! \NT".
26. In Supplementary Figure 6, write the values in circle graphs and bar graphs.
27. At Line 353, \As shown, very low levels of PUFA-containing PC were observed under these conditions, in clear contrast to what is observed in muscles (Fig.1)", what are \these conditions". This is true for NT and 16:0, however, the levels of PUFA-containing PC is not so low for the other conditions.
28. At Line 360, SFA and MUFA rst appear in the manuscript. Explain them. (I guess they mean saturated fatty acid and monounsaturated fatty acid.)
29. From Line 360 to 367, the explanation about EFA and production of AA and DHA should be mentioned in Introduction.
30. From Line 373 to 378, why 18:2 and 18:3 did not lead to the formation of AA- and DHA-containing PC. In addition, although the authors claim so that, the fractions of PC with DB > 2 for 18:2 is the almost same as that for 22:6 and that for 18:3 is larger than that for 22:6 as shown in Supplementary Figure 6.
31. At Line 377, the data for PE, PS, and PI can be shown as Supprementary Figures.
32. In Fig.5, add observation time and scale bar in Fig.5E, add axes instead of scale in Fig.5F, and what is F0 in Fig.5F?
33. In the caption of Fig.5, write the value of the scale bar in Fig.5B.
34. At Line 388, as a function of time (F). Amplitude ! as a function of time. (F)
Amplitude
35. In Fig.6D, the result shows which condition (cell)?
36. At Line 405, what is the reference fluorescence intensity of \Relative
fluorescence intensity" (how did the authors normalize the fluorescence intensity?).
37. At Line 406, how did the authors determine 10 % for the threshold percentage of breakage?
38. In Sec.3.3, the cells are immersed in a hypotonic solution, increasing the volume of the cells. I do not understand why it causes deformation. The excess area required for deformation is usually determined by the membrane area and intracellular volume. If water enters the cell due to osmotic pressure and the intracellular volume increases, the excess area will decrease and deformation will be less likely to occur. This is usually accepted in lipid vesicle/liposome systems which is very simpler system than cells. (For example, U. Seifert, Advances in Physics, 46, 13-137 (1997).) Explain why
the deformation occurs even though water gets inside the cell and increases in volume. And, the authors wrote \osmotic-induced cell shrinking (19,20)." at Line 434. Why does the cell shrink even in a hypotonic solution?
39. In Sec.3.3, the authors should explain the formation mechanism of VLD. In addition, in my understanding, the intracelluar volume is very important to evaluate the deformation. How about measuring the intracelluar volume at least before and after deformation from a three-dimensional image of a confocal microscope?
40. At Line 464, Vmax is the part where the slope of the graph is the largest. Is this correct as an analysis method? Why don't you fit the fluorescence profile with a sigmoid function and discuss based on the slope of the function.
41. In Sec.3.4, the permeation of fluorescent probe (FM1-43) was observed. Interestingly, the permeation is frequently observed for non-treated myoblasts than PUFA-treated cells. The authors seem to think that the reason is the existence of VLDs, but the rationale is unclear. Why can the authors deny the possibility that the presence of PUFAs in the cell membranes improves permeability? Although related to my 35th
comment, each of non-treated cells and PUFA-treated cells can show
fluorescence intensity according to the number of VLDs.
42. In Fig.7B 3, to repeat the same comment, the intracellular volume suddenly decreased due to the formation of VLDs. Why does the volume decrease even for the cells in a hypotonic solution.
43. At Line 552, although the authors wrote that POPC was used, I cannot find the results for POPC in Fig.8. Because POPC is the most common molecule in cell membrane, the results of POPC and PUPC with POPC are the most important.
44. From Line 570 to 572, \This redistribution comes primarily for the readily available membrane reservoir, or buffer, present at the surface of the cell in the form of membrane superstructures, including VLDs (23,24).", this sentence completely confuses me. I thought that VLD would be formed as a result of membrane deformation, but this statement says that VLD is used to cause membrane deformation.
45. At Line 575, Fig.8B to Fig.8C.
46. In Fig.8, values and characters are too small to read. Especially, I cannot read values in G.
47. In Fig.8D-F, change the scale on the horizontal axis to the same as C.
48. In Fig.8D, why did not the authors show the results for DOPC and PAPC?
49. In Fig.8E and F, why did not the authors show the results for DPPC and DOPC?
50. In Fig.8, when the membrane is expanded, the molecules deform to secure the area. The results show that molecules like DHA and AA are spreading their tails. In Fig.8E, the PAPC with DOPC/PAPC is slightly wider than the PAPC with DPPC/PAPC. This result is counterintuitive because DOPC is a molecule that is easier to spread than DPPC. The similar behavior is also seen in PUPA in Fig.8F. Explain this behavior.
51. In Fig.8, were the collapse of the membrane structure, the void formation on membrane, the escape of lipid molecules from the membrane, and so on observed during the expansion of the membrane?
52. In Fig.8F, DHA chain in PUPC is relatively positionaed in parallel to the membrane plane. On the other hand, it is perpendicular to the membrane plane in PUPC single component membrane as shown in Fig.8D. Although this difference may arise from the interaction between PUPC and DPPC/DOPC, the authors did not mention about this point.
53. In Fig.8, the angle for PAPC in PAPC/DOPC in Fig.8E is quite different from that for PUPC in PUPC/DOPC in Fig.8F. Explain why the surface tension is the almost same as shown in Fig.8B.
54. In Fig.8, it is important to examine the behavior of PUPC and PAPC mixed membrane. It will be interesting to find out if PUPC and PAPC spred in the same way, or if only PUPC spreads selectively in future.
55. At Line 643, the authors showed a compact conformation of DHA. The increase of angle in DHA to 90 degree is thought to be the replenishment of the membrane area by expanding the membrane. However, the compact conformation does not seem to contribute significantly to replenishing the area. Thus, the DHA with the compact conformation does not need to take 90 degree. Why does the DHA take 90 degree even for the compact conformation?
56. At Line 720, (Fig.E-F) to (Fig.8E-F)
57. At Line 721, (Fig.D-F) to (Fig.8D-F)
Author Response
Reviewer 1:
Minor Comments
Minor comments 1, 2, 7, 8, 9, 10, 12, 13, 18, 25, 28, 45, 56, and 57, corresponding to typographical errors have been addressed.
Major Comments:
At Line 23, the authors should add some references.
The following references have been added to the text: (Antonny et al., 2015; Bazinet and Layé, 2014)
At Line 34, the authors should explain about Ref.[5] concretely, not just cited.
The explication of this reference is the object of the three paragraphs following Line 34:
“A very simple way to get a rapid overview of this fatty acid distribution is to consider the Double-Bond (DB) index of a given lipid class, which corresponds to the relative percentage of saturated (DB=0: i.e PL species containing no double-bond within their FA chains) versus monounsaturated (DB=1: one double bond), diunsaturated (DB=2: two double bonds) and polyunsaturated (DB > 2: > two double bonds) species within this lipid class. In rats, PC is the lipid class which displays the highest diversity in terms of its fatty acyl chain composition depending on the organ considered. When considering PC, rat organs display very characteristic fatty acid distributions within PLs, presenting various saturation rates: they can be classified as DB=0 (Spleen, lung), DB=1 (Brain), DB=2 (Pancreas) and DB>2 organs (liver, muscle and cardiovascular system) (Bacle et al., 2020).
Another intriguing observation is that, among PUFA, very specific signatures can be observed among organs when considering the relative percentages of ω-3 versus ω-6 PUFA. The main PUFA species found within PLs are the ω-6 Arachidonic Acid (AA) and the ω-3 Docohexaenoic Acid (DHA). In rats, one can differentiate AA- (liver and Cardiovascular system) and DHA-enriched (skeletal muscle) organs (Bacle et al., 2020).
Finally, the various organs do not respond similarly to selective diets. Saturated and Monounsaturated FA originating from a high-fat diet preferentially accumulate in rats within PC in their liver and skeletal muscles, in a process that occurs at the expense of PUFA. Interestingly, this FA redistribution is paralleled by a global decrease in muscle strength (Bacle et al., 2020).”
To help the reader make the connection between Ref. 5 and this description, and as recommended by Reviewer 2, we removed the paragraph break before “A very simple way…”.
From Line 36 to 55, the authors explain based on only Ref.[5]. Since Ref.[5] is the paper by the author's group, the authors should introduce the research background and motivation with comparing the other group studies.
The following references have been added to the text:
1) “organs display very characteristic fatty acid signatures” and “among PUFAs, very specific signatures can be observed among organs when considering the relative percentages of ω-3 versus ω-6 PUFAs”: (Harayama et al., 2014; Hicks et al., 2006)
2) “Various organs do not respond similarly to selective diets”: (Abbott et al., 2010; Rong et al., 2015)
At Line 60, the research motivation that PUFAs are included in muscle cells is not sufficient. Explain about the research motivation more carefully. I cannot understand why the authors think \DHA-containing PLs could be central players for muscle cell adaptation to mechanical constraints.".
The following sentence has been added to the text:
“Recent experiments and simulations suggest that PUFA-containing PLs respond differently to mechanical stress than other PLs, owing to their unique conformational plasticity (Manni et al., 2018; Pinot et al., 2014).”
11. At Line 199, 15 is a relative permittivity? If so, why was this value used. In general,
this value is too small for water and too high for hydrocarbon region.
In MARTINI, electrostatic interactions are evaluated only on charged beads representing chemical groups with a net charge (+2, +1, -1, -2, etc…, for beads representing ions, phosphates, cholines, etc...). The parameterization of the force field has been done by taking into account a default explicit screening by setting the relative permittivity epsilon of 15. So, one has to stick to this value when using MARTINI.
14. At Line 212, what is g?
The Greek symbols have disappeared in this version of the manuscript and we apologize for this. g is in fact γ, which corresponds to the surface tension in mN.m-1.
The text has been completed/corrected as follows:
“For all simulations, surface tension (γ) was computed from the diagonal values of the pressure tensor (Pxx, Pyy, and Pzz) using the Kirkwood-Irving method (Irving and Kirkwood, 1950): because of high fluctuations in Pxx, Pyy, and Pzz, γ fluctuates vigorously on a microscopic system of a few thousands of atoms such as ours. Thus, two simulations of the same system with constant xy total area but with different initial velocities can lead to different values of g on the hundreds of ns timescale. For this reason, the average and error on γ were evaluated from the g_energy GROMACS tool (g_energy outputs to the screen average and error using points at all time steps, which is more precise than generating an xvg file and doing the analysis on the latter) using the whole trajectory for each simulation.”
Since the authors discussed the PC species distribution based on the number of double bond, I suggest that the order of lipids is sorted by the number of double bond in Fig.1. Otherwise, display using different colors depending on the number of double bond.
As suggested by reviewer 1, we have tested different representations during the course of the revision process, including reordering of the data and the usage of different colors. However, in our opinion, these representations appeared more confusing than the present one and therefore, we decided to keep the original version of the Figure, which was already used elsewhere (Bacle et al., 2020), in this revised manuscript.
In Fig.2, values should be shown in circle graphs and bar graphs.
Values have been added to the Figure and the following sentence has been added to the corresponding legend: “The percentages of the various subspecies are indicated.”
In Fig.2, how did the authors obtain the AA/DHA ratio?
The following sentence explaining the method used has been added to the legend of Fig. 2:
“The ratio of DHA (PC 36:6 38:6 and 40:6) - to AA (PC 36:4, 38:4 and 40:4)-containing PC subspecies in the various organs is also displayed, and was calculated from the values displayed in Fig. 1.”
In Fig.3, the values in B and C are too small to read.
As suggested by Reviewer 2, panels B and C have been removed.
At Line 306, the authors wrote \If Arachidonic Acid (AA; 20:4) appeared as the most represented fatty acid...". What is the basis for thinking this way?
The corresponding sentence has been corrected as follows:
“If AA appeared as the most represented fatty acid in brain, lung and pancreas (see histograms in Fig. 2), as combinations with palmitate (PC 36:4) or stearate (PC 38:4; Fig. 1), DHA was exquisitely enriched in skeletal muscles and the heart, in combination with palmitate (PC 38:6) or stearate (PC 40:6; Fig.1).”
At Line 311, why did not the authors mention about PI here?
PI is mentioned in a previous paragraph of the same section: “As already described in previous studies (Hicks et al., 2006), it was observed that PC is the PL species which displays the widest variations in terms of fatty acid chain distribution depending on the organ considered (Fig. 1 and Supplementary Fig. 1), whereas PI is essentially represented by the species PI 38:4 in all the organs studied (Supplementary Fig. 2).”
At Line 319, the sentence of \PS fatty acid distributions could not ...." should appear also in the caption of Supplementary Figure 4.
The relevant sentence has been added to the legend of Supplementary Figure 4.
At Line 338, what is the important mechanical constraints. Explain more concretely.
The sentence has been modified as follows : “Because muscle cells follow repeated cycles of elongation and relaxation, muscle cells are constantly exposed to important mechanical constraints due to intrinsic repetitive muscle deformations (Burkholder, 2007) and are prone to experience increases in membrane tension (Roux et al., 2019)”
- At Line 340, whether does this \volume" mean membrane volume or cell volume?
It has been demonstrated that a limited degree of swelling occurs in muscle skeletal muscle during intense exercise and that this volume must return to its normal value during recovery.
Hence, it means cell volume.
In Supplementary Figure 6, write the values in circle graphs and bar graphs.
Values have been added to the Figure and the following sentence has been added to the corresponding legend: “The percentages of the various subspecies are indicated.”
At Line 353, \As shown, very low levels of PUFA-containing PC were observed under these conditions, in clear contrast to what is observed in muscles (Fig.1)", what are \these conditions". This is true for NT and 16:0, however, the levels of PUFA-containing PC is not so low for the other conditions.
We refer to the NT conditions, as mentioned in the text:
“First, C2C12 myoblasts were cultivated under usual conditions (no fatty acid supplementation; Non-Treated: NT), and the fatty acyl distribution within PC was determined by ESI-MS as described under “Materials and Methods”. The results are displayed as radar graphs in Fig. 4 and as histograms in Supplementary Fig. 5, and the corresponding DB indexes are presented in Supplementary Fig. 6. As shown, very low levels of PUFA-containing PC were observed under these conditions, in clear contrast to what is observed in muscles (Fig. 1).”
From Line 360 to 367, the explanation about EFA and production of AA and DHA should be mentioned in Introduction.
and
30. From Line 373 to 378, why 18:2 and 18:3 did not lead to the formation of AA- and DHA-containing PC.
Including the paragraph about the way AA and DHA are synthesized from EFA (i. e. 18:2 and 18:3) in the Results rather than in the Introduction section, was done on purpose to allow a direct connection between this process and the absence of an efficient transformation of these precursors to their end products under usual in vitro experimental conditions:
“By contrast to saturated fatty acids (SFAs) and monounsaturated fatty acids (MUFAs), the body cannot synthesize PUFA “from scratch” and must therefore obtain basic blocks from the diet, which are referred to as Essential Fatty Acids (EFA). As already mentioned, there are two families of EFA, which are classified on the basis of the position of the first double-bond from the last carbon of the acyl chain (the omega (ω) carbon): ω -3 and ω-6. The ω-6 building block is linoleic acid (18:2) and the ω -3’s is α-linolenic acid (18:3). These precursors are metabolized by a series of alternating desaturation and elongation reactions to form the main end-products AA (ω -6) and DHA (ω -3) (Al-Turkmani et al., 2007). End-PUFA products can also be directly obtained from the diet. For example, DHA is present in fish oils. Therefore, C2C12 myoblasts were cultivated in the presence of 100 µM of either the end-products AA and DHA, or their precursors 18:2 and 18:3, and the distribution of these fatty acids within PC species was evaluated by ESI-MS (Fig. 4). As shown, AA addition resulted in the formation of the AA-containing PC species PC 36:4 and PC 38:4, showing that this FA was efficiently incorporated within PLs. Similarly, incubation with DHA induced the production of the DHA-containing species PC 38:6, which is the most prominent species in muscles (Fig. 1). Interestingly, incubation with 18:2 and 18:3 did not lead to the optimal formation of the corresponding AA- and DHA-containing PC species, suggesting that these precursors were not efficiently subjected to the elongation/desaturation steps required to form the corresponding end-products in C2C12 myoblasts, at least efficiently enough to reproduce the lipid signatures observed in vivo (Fig. 1).”
- In addition, although the authors claim so that, the fractions of PC with DB > 2 for 18:2 is the almost same as that for 22:6 and that for 18:3 is larger than that for 22:6 as shown in Supplementary Figure 6.
We agree with Reviewer 1. Claiming that incubation with 18:2 and 18:3 did not lead to the formation of the corresponding AA- and DHA-containing PC species is excessive and oversimplified. We modified the corresponding sentence as follows:
“Interestingly, incubation with 18:2 and 18:3 did not lead to the optimal formation of the corresponding AA- and DHA-containing PC species, suggesting that these precursors were not efficiently subjected to the elongation/desaturation steps required to form the corresponding end-products in C2C12 myoblasts, at least efficiently enough to reproduce the lipid signatures observed in vivo (Fig. 1).”
At Line 377, the data for PE, PS, and PI can be shown as Supplementary Figures.
To stay in line with the comments from Reviewer 2 who underlined that the amount of data presented in this manuscript is somehow overwhelming, making it difficult for the reader to get the main points, we preferred not to add these data, which do not bring much to the story and which have required the addition of 3 supplementary Figures.
In Fig.5, add observation time and scale bar in Fig.5E, add axes instead of scale in Fig.5F, and what is F0 in Fig.5F?
Fig. 5F has been modified as suggested; F0 is the basal fluorescence level before the apparition of the VLD.
In the caption of Fig.5, write the value of the scale bar in Fig.5B.
Added: scale bar is 5 µm
34. At Line 388, as a function of time (F). Amplitude ! as a function of time. (F)
Amplitude
“Amplitude” has been replaced by “Plateau”
In Fig.6D, the result shows which condition (cell)?
This data displayed correspond to the results from all conditions, showing a clear-cut relationship between the increase in fluorescence and the number of VLDs, whatever the condition tested. The sentence has been modified as follows: “(D) Maximal relative fluorescence obtained in labeled-cells in all conditions as a function of the number of VLDs.”
At Line 405, what is the reference fluorescence intensity of \Relative
fluorescence intensity" (how did the authors normalize the fluorescence intensity?).
Relative fluorescence intensity is relative to basal fluorescence before osmotic shock. Sentence has been modified as follows: “Fluorescence intensity, relative to basal, obtained at the end of osmotic downshock.”
At Line 406, how did the authors determine 10 % for the threshold percentage of breakage?
This is not 10 % but 10 of relative fluorescence.
The sentence has been modified as follows: “The % indicated in this figure corresponds to the percentages of C2C12 exhibiting relative fluorescence intensities above 10, which corresponds to bona fine cell breakage, under the various culture conditions.”
- In Sec.3.3, the cells are immersed in a hypotonic solution, increasing the volume of the cells. I do not understand why it causes deformation. The excess area required for deformation is usually determined by the membrane area and intracellular volume. If water enters the cell due to osmotic pressure and the intracellular volume increases, the excess area will decrease and deformation will be less likely to occur. This is usually accepted in lipid vesicle/liposome systems which is very simpler system than cells. (For example, U. Seifert, Advances in Physics, 46, 13-137 (1997).) Explain why
the deformation occurs even though water gets inside the cell and increases in volume. And, the authors wrote \osmotic-induced cell shrinking (19,20)." at Line 434. Why does the cell shrink even in a hypotonic solution?
The behavior you describe is indeed what is observed in many cells unable to form VLDs. In the case of muscle cells or neurons, discrete sites of invagination, sites of nascent VLDs, are present on the adhesive part of the cell to the substrate. It has been hypothesized that the flowing bilayer of nascent VLDs may be inordinately water permeable (water permeability increases with membrane fluidity (Lande et al., 1995) and particularly with PUFA-containing membranes (Manni et al., 2018)). If so, the increasing water can exit the cell and fill the new compartment inside the VLDs at an abnormally high rate, creating hydrostatic pressure that further increases VLD bilayer tension, causing additional flow of bilayer to the invagination (Morris and Homann, 2001). Hence, even if osmotic downshock occurs, inducing an increase in cell volume, VLDs formation allows water efflux inside the VLD and then cell shrinking. In DHA conditions, the elevated number of VLDs combined with an increased water permeability related to the presence of a high percentage of PUFA-containing PLs within the plasma-membrane, may account for the observed reduction in cell volume, in a process that could be particularly efficient in preventing cell breakage under osmotic downshock. This is a very exciting hypothesis, which will be the object of future studies from our groups.
In Sec.3.3, the authors should explain the formation mechanism of VLD. In addition, in my understanding, the intracelluar volume is very important to evaluate the deformation. How about measuring the intracelluar volume at least before and after deformation from a three-dimensional image of a confocal microscope?
The following sentence has been added in Sec 3.3: “It has been hypothesized that, during osmotic downshock, the flowing bilayer of nascent VLDs may be water permeable and increasing intracellular water could exit the cell and fill VLDs at a high rate (Morris and Homan, 2001).”
Measuring the intracellular volume could be performed through kinetics 3D images of cells loaded with a vital fluorescent dye, and it would indeed generate interesting data.
However, in this study, we focused on cell surface variations because this parameter is directly connected to the process of interest, i. e. phospholipid adaptation to membrane extension under cell swelling. However, we describe in lines 169 to 177 how this parameter can reflect global volume variations:
“Since myoblasts can be considered as hemispheres, variations in the area of a given cell, provided that the distance between this given section and the base of the hemi-sphere remains constant (which is guaranteed by the fact that cells are adhered to the support), are directly proportional to surface variations of the overall hemisphere. The initial surface was determined before osmotic downshock. When maximum surface was reached following this treatment (Surface max), the surface of the cross section was determined and the corresponding time was noted (tSurface max). When cell breakage was visualized using transmission under a given condition following osmotic downshock, the time when this breakage occurred was also determined (t breakage).”
We believe that surface variations, which are much easier to determine automatically than volume using this method for an important number of cells under the various conditions, is pertinent to evaluate cell deformation during the osmotic downshock experiments performed in this study.
At Line 464, Vmax is the part where the slope of the graph is the largest. Is this correct as an analysis method? Why don't you fit the fluorescence profile with a sigmoid function and discuss based on the slope of the function.
We initially planned to perform the analyses according to the method you propose, but sigmoid curves display various fitting parameters that we were not able to obtain automatically using usual software, such as GraphPad Prism. We then chose Vmax, the maximal velocity of VLDs formation (obtained automatically with the maximum value of the derivative) and tVmax, which corresponds to the time when Vmax is reached, as potent descriptors of the kinetics of VLD formation.
In Sec.3.4, the permeation of fluorescent probe (FM1-43) was observed. Interestingly, the permeation is frequently observed for non-treated myoblasts than PUFA-treated cells. The authors seem to think that the reason is the existence of VLDs, but the rationale is unclear. Why can the authors deny the possibility that the presence of PUFAs in the cell membranes improves permeability? Although related to my 35th
comment, each of non-treated cells and PUFA-treated cells can show
fluorescence intensity according to the number of VLDs.
The FM1-43 probe exhibits a higher fluorescence yield in the cytoplasm at 543 nm. This property has been used to evaluate membrane ruptures (Cai et al., 2009), from which the probe enters the cell, leading to a global increase in fluorescence intensity. In our experiments, membrane ruptures appeared to occur more in cells that are less capable of producing VLDs. This is particularly the case for NT cells, while PUFAs treated cells, which produce VLDs at high rates, exhibit lower intracellular fluorescent intensities, due to a reduction in membrane rupture.
In Fig.7B 3, to repeat the same comment, the intracellular volume suddenly decreased due to the formation of VLDs. Why does the volume decrease even for the cells in a hypotonic solution.
The volume decreases as described in the response to point 38.
43. At Line 552, although the authors wrote that POPC was used, I cannot find the results for POPC in Fig.8. Because POPC is the most common molecule in cell membrane, the results of POPC and PUPC with POPC are the most important.
In all the simulations performed during the course of this study, membranes bearing POPC behaved, as expected, in an intermediate way between DPPC- and DOPC- containing membranes. As mentioned by Reviewer 1, we performed an enormous number of simulations that cannot all be displayed in this manuscript. We therefore chose in Figure 8 to display representative experiments, corresponding to the most “extreme” behaviors (for example PAPC and PUPC in DPPC or DOPC, Fig. E-F). As Reviewer 1 can see in Figure 8 D-F, the behavior of the palmitoyl chain is largely unaffected by its neighboring chain (either fully saturated (DPPC) or polyunsaturated (PUPC); Fig. D) and AA and DHA reactions to extension are largely independent of the nature of the surrounding PL (either DPPC or DOPC; Fig. 8. E and F). The exact same behaviors were observed when POPC was used. The general lesson from these simulations is that the response of an acyl chain to surface extension is related to its own intrinsic properties rather than to the nature of its acyl chain neighbors, either in the same PL or in the surrounding ones. This behavior can clearly be seen with the data displayed in Fig.8. Adding the data corresponding to POPC to Fig. D-F results in line superpositions, making the overall data impossible to visualize correctly. We therefore decided not to add these data to the present manuscript.
The following sentence was added to the text: “Since POPC membranes systematically displayed an intermediate behavior between DPPC and DOPC bilayers in all the simulations performed in this study, we chose not to show the corresponding data to simplify the reading.”
Concerning the impacts of PUFA on surface tension (Fig. 8B), we added the following sentence to the text:
“However, even if the presence of PUFA in PC species seemed to slightly decrease surface tension under surface expansion as compared to pure DOPC bilayers, no significant variations were observed, suggesting that the presence of PUFA, either AA (PAPC) or DHA (PUPC), are not strong reducers of membrane tension per se. The exact same behaviors were observed when DOPC was replaced by either DPPC or POPC (data not shown).”
- From Line 570 to 572, \This redistribution comes primarily for the readily available membrane reservoir, or buffer, present at the surface of the cell in the form of membrane superstructures, including VLDs (23,24).", this sentence completely confuses me. I thought that VLD would be formed as a result of membrane deformation, but this statement says that VLD is used to cause membrane deformation.
According to the mechanism previously described, VLDs result from membrane tension and water influx. Hence, portions of membranes that constitute VLDs are considered as membrane reservoir to compensate and reduce membrane tension of the whole cell.
In Fig.8, values and characters are too small to read. Especially, I cannot read values in G.
This modification has been done.
In Fig.8D-F, change the scale on the horizontal axis to the same as C.
This modification has been done.
In Fig.8D, why did not the authors show the results for DOPC and PAPC?
Please, see point 43.
In Fig.8E and F, why did not the authors show the results for DPPC and DOPC?
Please, see point 43.
In Fig.8, when the membrane is expanded, the molecules deform to secure the area. The results show that molecules like DHA and AA are spreading their tails. In Fig.8E, the PAPC with DOPC/PAPC is slightly wider than the PAPC with DPPC/PAPC. This result is counterintuitive because DOPC is a molecule that is easier to spread than DPPC. The similar behavior is also seen in PUPA in Fig.8F. Explain this behavior.
As Reviewer 1 could notice, the differences in AA and DHA spreading depending on the nature of the surrounding PL (DPPC or DOPC) are very small. This shows, as mentioned in point 43, that PUFA reaction to surface expansion is poorly influenced by the acyl chain content of the neighboring PLs. However, it appears indeed that AA and DHA tend to slightly prefer the extended conformation when in the presence of DOPC rather than in the presence of DPPC, particularly at the lowest membrane extension values (Equil and 5 % extension). This is likely due to the fact that the kinks found in the oleyl chains of DOPC produce deep surface voids (Antonny et al., 2015), therefore generating more “room” for the PUFA chain to expand in the plane of the bilayer.
In Fig.8, were the collapse of the membrane structure, the void formation on membrane, the escape of lipid molecules from the membrane, and so on observed during the expansion of the membrane?
We could indeed observe pore formations but for extensions > 25 %. For example, pore formation could be observed for a surface extension of + 31% with a pure DOPC bilayer. Interestingly, such extreme extension values were not observed with C2C12 cells under osmotic downshock, whatever their membrane composition (Figure 6E). However, we preferred not to include such a discussion in the present manuscript to avoid an overinterpretation of the data.
In Fig.8F, DHA chain in PUPC is relatively positionaed in parallel to the membrane plane. On the other hand, it is perpendicular to the membrane plane in PUPC single component membrane as shown in Fig.8D. Although this difference may arise from the interaction between PUPC and DPPC/DOPC, the authors did not mention about this point.
Fig. 8D shows the distribution of the angle alpha for the sn-1 Palmitoyl chain (not DHA) either in DPPC or PUPC membranes. The goal of this simulation was to check whether the presence of a PUFA could influence the propension of the palmitoyl chain to expand in the plane of the bilayer under surface extension, which is not the case.
“As shown in Fig. 8D, when considering a saturated fatty acyl chain such as Palmitate, at the sn-1 position of PC, the distribution of the angle α to be quite restricted to values around 30° (or 180-30=150° for same PC species distributed in the opposite leaflet of the membrane bilayer; Fig. 8C) at equilibrium (i. e. when no extension was applied). This distribution was very similar whether another Palmitate chain (DPPC) or a PUFA such as DHA (PUPC) occupied the sn-2 position.”
In Fig.8, the angle for PAPC in PAPC/DOPC in Fig.8E is quite different from that for PUPC in PUPC/DOPC in Fig.8F. Explain why the surface tension is the almost same as shown in Fig.8B.
We indeed expected DHA, considering its ability to expand in the plane of the bilayer, to be a strong reducer of surface tension under membrane extension. However, this is not what we observed (Fig. 8B), at least using molecular dynamics simulations. I am afraid that we cannot provide a clear explanation for this observation at this step. Evaluating surface tension by experimental means using membranes of various compositions under extension could undoubtfully bring a more clear-cut answer to this very interesting question.
- In Fig.8, it is important to examine the behavior of PUPC and PAPC mixed membrane. It will be interesting to find out if PUPC and PAPC spred in the same way, or if only PUPC spreads selectively in future.
If very interesting on a basic point of view, such membranes composed of 100 % PUFA-containing PLs are not encountered in vivo (Fig. 2). This is why we did not consider studying the behavior of PUPC and PAPC mixed membranes as a priority in this study. However, these are simulations that we will very likely perform in the future to evaluate the impacts of AA on DHA expansion and vice-versa.
At Line 643, the authors showed a compact conformation of DHA. The increase of angle in DHA to 90 degree is thought to be the replenishment of the membrane area by expanding the membrane. However, the compact conformation does not seem to contribute significantly to replenishing the area. Thus, the DHA with the compact conformation does not need to take 90 degree. Why does the DHA take 90 degree even for the compact conformation?
This is again a very interesting question. As Reviewer 1 will notice on Fig. 8G, a +20% extension of the membrane tends to favor the expanded 90° conformation of DHA at the expense of the compact one, as compared to equilibrium, which is in good agreement with the notion that the DHA chain expands in the plane of the bilayer when the membrane is extended. The fact that the DHA chain adopts the compact 90° form under equilibrium (no extension) is a recurrent observation in the literature: even if somehow counterintuitive, this corresponds to the ability of DHA to bring its ω- carbon to the external face of the membrane, forming shallow surface defects (Pinot et al., 2014).
Reviewer 2 Report
The study is a valuable and interesting take on the role of polyunsaturated fatty acids (PUFA) in cellular membranes and its potential role as modulators of the impact of mechanical tensions on cell surface. However, the authors often in the results present comparisons with other animal models, which would be better suited for the discussion, and make the reading somewhat tedious by explaining methods that were previously detailed in the appropriate section. Overall, the construction of sentences should be revised to facilitate reading. There are also too many paragraph breaks in the “Introduction” and “Results”; the authors should group paragraphs dedicated to the same subject, and single phrase paragraphs should be avoided.
The authors present a very interesting set of results although the supplied information in figures seem at times excessive. Overall I have difficulties in appreciating the results presented because the message is not sufficiently focused. For instance, on figure 3, I fail to see the relevance of presenting the correlation coefficients of the DB indices to the components (panels B and C) when one can see their correlation already on panel A. If the authors wish to present this information, they should place it in the supplementary file. I fail do see the purpose of presenting an ellipse in panel A, and it would be more interesting to show the XY axes in the scale -3 to 3.
The names of lipid classes should not be presented in capital letters in the text, because it makes the text look somewhat clumsy and it impairs its reading flow. e.g. in line 19 it is written “PolyUnsaturated Fatty Acids (PUFA),” and I would recommend the use of the more elegant “polyunsaturated fatty acids (PUFA)” or the more common in scientific literature “poly-unsaturated fatty acids”. The same applies to other named compounds and organelles throughout the text. I would advise against the use of capital letters in the middle of sentences, generally.
The first time a FA is mentioned in the text, the authors should explain the acronym or its notation, e.g. "docosahexaenoic acid (DHA, 22:6w3)" and then use only DHA on the remaining instances it is used.
I revised the text until line 300 and I recommend the authors to revise the remaining text according to the recommendations and corrections presented.
Please revise the results to reduce the instances when unnecessary methodology is described (it is already described in “methods” section).
In the end of Introduction, the authors claim that (line 68) “observations suggest that DHA enrichment in these tissues may not be mere coincidence, but the result of a selection process during evolution to respond at best to the mechanical constraints muscles are confronted to.” I find it a little too strong to claim that the PL profile of muscles results from a selection process during evolution because no experiments were made in that sense. At the most the authors may claim that “DHA enrichment in muscle tissues may not be haphazard, but the result of muscle adaptation to imposed mechanical constraints.” (please note that is not appropriate to say that muscles are “confronted” to “mechanical constraints”).
Detailed comments – I have made numerous comments in the pdf file.
Abstract – line 8
“PUFA are particularly enriched in the membrane Phospholipids (PLs) of the cardiovascular system and skeletal muscles.”
I think the authors mean that “Membrane Phospholipids (PLs) of the cardiovascular system and skeletal muscles are particularly enriched in PUFA.”
Line 289
“Concerning PC, and as already reported for rats, it clearly appears that some organs are particularly enriched in species bearing polyunsaturated fatty acid chains (PUFA; more than two double-bonds/unsaturations in their fatty acid chains; e. g. PC 38:4), whereas others preferentially contain PC with two saturated fatty acyl chains (e. g. PC 32:0) (Fig. 1).”
I would recommend a revision of the text to a more focused form:
“As previously reported for rats (Reference!), it was observed that some organs are enriched in PC species with PUFA acyl chains (e.g. PC 38:4), whereas other organs preferentially contain PC with two saturated fatty acyl chains (e. g. PC 32:0)”
Line 294 – the text has an excess of explanation as should focus on the results of the analysis, e.g. the following sentence can be trimmed as follows:
“To visualize better these variations, the Double-Bond (DB) index was calculated in each case (Fig. 2). When performing a multivariate “Principal Component Analysis” (PCA) based on the DB index (Fig. 3), the striking differences between the various organs could be clearly visualized.”
should be re-written as
“These variations can be visualized by the DB index calculated for each organ (Fig. 2), especially when PCA is applied the index (Fig. 3).

Author Response
Reviewer 2:
- However, the authors often in the results present comparisons with other animal models, which would be better suited for the discussion, and make the reading somewhat tedious by explaining methods that were previously detailed in the appropriate section. Overall, the construction of sentences should be revised to facilitate reading. There are also too many paragraph breaks in the “Introduction” and “Results”; the authors should group paragraphs dedicated to the same subject, and single phrase paragraphs should be avoided.
We considered carefully Reviewer 2’s recommendations and hope that the present manuscript reads better.
- For instance, on figure 3, I fail to see the relevance of presenting the correlation coefficients of the DB indices to the components (panels B and C) when one can see their correlation already on panel A. If the authors wish to present this information, they should place it in the supplementary file. I fail do see the purpose of presenting an ellipse in panel A, and it would be more interesting to show the XY axes in the scale -3 to 3.
We considered Reviewer 2’s comment and modified the panel A of Fig. 3 as requested. The panels B and C have also been removed.
The Legend of the Figure has been modified as follows: “Principal Component Analyses (PCA) of the variability in PC DB index as a function of the organ. The variables measured in this study (i. e. DB=0, DB=1, DB=2 and DB>2) are expressed as vectors in dashed lines.”
- The names of lipid classes should not be presented in capital letters in the text, because it makes the text look somewhat clumsy and it impairs its reading flow. e.g. in line 19 it is written “PolyUnsaturated Fatty Acids (PUFA),” and I would recommend the use of the more elegant “polyunsaturated fatty acids (PUFA)” or the more common in scientific literature “poly-unsaturated fatty acids”. The same applies to other named compounds and organelles throughout the text. I would advise against the use of capital letters in the middle of sentences, generally.
This has been modified all along the manuscript.
- The first time a FA is mentioned in the text, the authors should explain the acronym or its notation, e.g. "docosahexaenoic acid (DHA, 22:6w3)" and then use only DHA on the remaining instances it is used.
This correction has been made.
- I revised the text until line 300 and I recommend the authors to revise the remaining text according to the recommendations and corrections presented.
Please revise the results to reduce the instances when unnecessary methodology is described (it is already described in “methods” section).
We modified the text considering Reviewer 2’s recommendations.
- In the end of Introduction, the authors claim that (line 68) “observations suggest that DHA enrichment in these tissues may not be mere coincidence, but the result of a selection process during evolution to respond at best to the mechanical constraints muscles are confronted to.” I find it a little too strong to claim that the PL profile of muscles results from a selection process during evolution because no experiments were made in that sense. At the most the authors may claim that “DHA enrichment in muscle tissues may not be haphazard, but the result of muscle adaptation to imposed mechanical constraints.” (please note that is not appropriate to say that muscles are “confronted” to “mechanical constraints”).
This change has been done.
- Detailed comments – I have made numerous comments in the pdf file.
The detailed comments from Reviever 2 have been included in the present version of the manuscript.
- Abstract – line 8
“PUFA are particularly enriched in the membrane Phospholipids (PLs) of the cardiovascular system and skeletal muscles.”
I think the authors mean that “Membrane Phospholipids (PLs) of the cardiovascular system and skeletal muscles are particularly enriched in PUFA.”
This change has been done
- Line 289: “Concerning PC, and as already reported for rats, it clearly appears that some organs are particularly enriched in species bearing polyunsaturated fatty acid chains (PUFA; more than two double-bonds/unsaturations in their fatty acid chains; e. g. PC 38:4), whereas others preferentially contain PC with two saturated fatty acyl chains (e. g. PC 32:0) (Fig. 1).”
I would recommend a revision of the text to a more focused form:
“As previously reported for rats (Reference!), it was observed that some organs are enriched in PC species with PUFA acyl chains (e.g. PC 38:4), whereas other organs preferentially contain PC with two saturated fatty acyl chains (e. g. PC 32:0)”
This modification has been made.
- Line 294 – the text has an excess of explanation as should focus on the results of the analysis, e.g. the following sentence can be trimmed as follows:
“To visualize better these variations, the Double-Bond (DB) index was calculated in each case (Fig. 2). When performing a multivariate “Principal Component Analysis” (PCA) based on the DB index (Fig. 3), the striking differences between the various organs could be clearly visualized.”
should be re-written as
“These variations can be visualized by the DB index calculated for each organ (Fig. 2), especially when PCA is applied the index (Fig. 3).
This modification has been made.
Round 2
Reviewer 1 Report
The authors revised the manuscript appropriately and I was satisfied with the modifications.
I recommend that the version will be published in Cells.
Author Response
We thank Reviewer 1 for recommending the publication of this version of our manuscript.
Reviewer 2 Report
I would like to thank the authors for the improvements made on the manuscript. Nevertheless, the authors only addressed the comments and suggestions made in the text and didn’t improve the sections that were not specifically edited. As I mentioned in the previous revision, the authors should revise the text according to suggestions made in the text and apply those suggestion to the WHOLE text. Take the example of sentence in line 440: “Therefore, it appeared from these experiments that the cellular PL composition influences the ability of the cells to form VLDs under osmotic stress.” This diminishes the importance of the results, the authors should write something along the lines “These results suggest that the cellular PL composition influences the ability of the cells to form VLDs under osmotic stress”. The captions of the figures that I did not edit previously were also not edited in this revised manuscript; they remain far too long and contain information that should be conveyed in methods. Please revise the text in its totality for casual sentences, and check which information should be conveyed under “methods”.
There is an interesting aspect that the authors dismiss a little. It concerns the lack of ability of the tested cells to bioconvert DHA and AA from the corresponding PUFA precursors. Many cells do not have, in fact, the ability to bioconvert certain PUFA to the corresponding longer chained PUFA. It would be really interesting if the authors offered some information on the cells they've used. Is it known that they have this ability, or lack of it? What about other muscle cells? Can muscle cells only incorporate the necessary PUFA from the diet, and do not have the ability to bioconvert it from other FA? This has massive repercussions for muscle health and human nutrition, and a small paragraph on it would be much appreciated.
Specific comments
I’ve made various suggestions throughout the text and I’m afraid I should insist on the FA notation. It is more precise to always state the nature of the unsaturation when it is stated that it exists. For instance, line 48, the authors state “The main PUFA species found within PLs are the ω-6 arachidonic acid (AA; 20:4) and the ω-3 docohexaenoic acid (DHA; 22:6).” To be precise, it should be written:
“The main PUFA species found within PLs are the ω-6 arachidonic acid (AA; 20:4ω6) and the ω-3 docohexaenoic acid (DHA; 22:6ω3).” The same applies to other places in the text, especially when the type of ω unsaturation is of relevance, please revise e.g. line 105: “Linoleic Acid (18:2), linolenic acid (18:3), AA and DHA (Santa Cruz Biotechnology, 105 Dallas, USA), (…)” should be written:
“Linoleic acid (18:2ω6), linolenic acid (18:3ω3), AA and DHA (Santa Cruz Biotechnology, 105 Dallas, USA), (…)”
Please remember that your focus is on the nature of unsaturations, it is important for the reader to have a clear idea of what is ω6 or ω3.
In figure 2, please align “Muscle” with the other charts.
Line 309
I would remove the “therefore” in the sentence. Please consider that you can have different FA assemblages with the same unsaturation rate. "therefore" implies that "various saturation rates" derive from the previous information, and you cannot conclude that. More, with your BD scheme, you mention unsaturation not saturation. Please correct.
Line 317
Please don’t start a paragraph with “because”. You may give more emphasis to what muscle cells do as “Muscle cells undergo repeated cycles of elongation and relaxation, and are thus exposed to important mechanical constraints due to intrinsic repetitive muscle deformations (..)”
Line 328-360
The authors give information that belongs to the methods. The authors should present the results and then reference the figures and supplement material. Even if the authors wish to present the method of cultivation here, they must present first the method and then the results. When describing the results, then they should refer the figures (Fig. 4, etc). The text must be revised here.
Figure 5. The information on the caption should be succinct. I would advise to transfer the mode of image capture to the methods section. The same applies to caption of figure 6 that I find unnecessarily long.
Line 360
The authors mention “behaviors” of PL regarding the FA pattern of their acyl chains, which is not correct. The cells may have a behavioural pattern of incorporation, not the PLs. The sentence “Similar behaviors were observed with other PLs, namely PE, PS and PI (data not-shown).” when in fact what they observe as “The lack of AA and DHA incorporation into PC was also observed for the other PLs species PE, PS and PI (data not shown).”
Line 403-405 is just lost in the text. I don’t even know what the “therefore” means here. I suppose it follows the sentence ending in line 360, it should NOT be a different paragraph. I sound repetitive, but please ask yourselves what is the rule that you follow to make a new paragraph.
Lines 406-417 are methods again. Some information is already given under “2.5. Cell labelling for confocal microscopy and methods for image analyses”, why not present the full description of methods in that section?
As I mentioned in the previous revision, the authors should revise the text according to suggestions made in the text and apply those suggestion to the WHOLE text. It is not the job of the reviewer to edit the text sentence by sentence, the authors should do that. Take the example of sentence in line 440:
“Therefore, it appeared from these experiments that the cellular PL composition influences the ability of the cells to form VLDs under osmotic stress.” This is just messy, you should write something along the lines “These results suggest that the cellular PL composition influences the ability of the cells to form VLDs under osmotic stress”. I apologise for being blunt but “appeared from this experiments” is not a good way to convey the information that results point in a certain direction, it makes your results “appear” as casual observations, which they are not.
Please revise the remaining text, and check which information should be conveyed under “methods”.

Author Response
Preliminary remarks
We thank again Reviewer 2 for his helpful comments. We tried very hard to fulfill his expectations in this new version.
We agree that some information that appeared in the “Results” or in ”Figure legends” could be conveyed to the “Materials ad Methods” section, and the present version was carefully edited in this way. For example, the legends of Fig. 5 and 6 have been drastically reduced.
However, it was not systematically possible, because, as highlighted by both reviewers, we used an important number of methods during the course of this study, and it appeared important for us to keep some summaries of the “Materials ad Methods” in the “Results” sections when needed, to avoid the reader to shift systematically back and forth between various parts of the manuscript, which may have even complexified the reading. For example, we decided to keep the legend of Fig. 8 under its original format. Among the 57 points raised by Reviewer 1 during the previous review round, which we all addressed, some of them consisted in reinserting some method details in the “Results” section, with the argument that it may facilitate the reading… We hope that we have reached a correct balance in this new version of the manuscript.
Reviewer 2 points that “the authors should revise the text according to suggestions made in the text and apply those suggestion to the WHOLE text. It is not the job of the reviewer to edit the text sentence by sentence, the authors should do that.” We fully agree with Reviewer 2 that it is not the job of the reviewer to edit the text sentence by sentence. However, we are quite confident that our manuscript is, as a whole, grammatically correct. Accordingly, neither Reviewer 1 nor the editor requested the manuscript to be edited for the English language. In this context, most of the amendments to the text that are proposed (or expected) are stylistic changes, rather than objective grammatical corrections. Such changes are very subjective, and it is very difficult for an author to anticipate what could be considered by another person a better way to formulate a given idea. Therefore, we did our best to see the text through the prism of Reviewer 2, based on some of his explicit examples, and we hope that it is now more conform to his expectations.
Point by point response to Reviewer 2’s comments
- I would like to thank the authors for the improvements made on the manuscript. Nevertheless, the authors only addressed the comments and suggestions made in the text and didn’t improve the sections that were not specifically edited. As I mentioned in the previous revision, the authors should revise the text according to suggestions made in the text and apply those suggestion to the WHOLE text. Take the example of sentence in line 440: “Therefore, it appeared from these experiments that the cellular PL composition influences the ability of the cells to form VLDs under osmotic stress.” This diminishes the importance of the results, the authors should write something along the lines “These results suggest that the cellular PL composition influences the ability of the cells to form VLDs under osmotic stress”. The captions of the figures that I did not edit previously were also not edited in this revised manuscript; they remain far too long and contain information that should be conveyed in methods. Please revise the text in its totality for casual sentences, and check which information should be conveyed under “methods”.
Please see “Preliminary remarks”. The sentence in line 440 has been modified as suggested by Reviewer 2.
- There is an interesting aspect that the authors dismiss a little. It concerns the lack of ability of the tested cells to bioconvert DHA and AA from the corresponding PUFA precursors. Many cells do not have, in fact, the ability to bioconvert certain PUFA to the corresponding longer chained PUFA. It would be really interesting if the authors offered some information on the cells they've used. Is it known that they have this ability, or lack of it? What about other muscle cells? Can muscle cells only incorporate the necessary PUFA from the diet, and do not have the ability to bioconvert it from other FA? This has massive repercussions for muscle health and human nutrition, and a small paragraph on it would be much appreciated.
The following paragraph has been added to the “Discussion” section (l. 659):
“Such a low conversion efficiency of EFA precursors to their final products has been observed in vivo (43). For example, the conversion of ingested 18:3ω-3 to DHA is very inefficient in Man, with <10% conversion in women and <3% in men (44, 45). This is likely due to the fact that a specific desaturase, namely ∆6 desaturase, is rate limiting (46).
- I’ve made various suggestions throughout the text and I’m afraid I should insist on the FA notation. It is more precise to always state the nature of the unsaturation when it is stated that it exists. For instance, line 48, the authors state “The main PUFA species found within PLs are the ω-6 arachidonic acid (AA; 20:4) and the ω-3 docohexaenoic acid (DHA; 22:6).” To be precise, it should be written:
“The main PUFA species found within PLs are the ω-6 arachidonic acid (AA; 20:4ω6) and the ω-3 docohexaenoic acid (DHA; 22:6ω3).” The same applies to other places in the text, especially when the type of ω unsaturation is of relevance, please revise e.g. line 105: “Linoleic Acid (18:2), linolenic acid (18:3), AA and DHA (Santa Cruz Biotechnology, 105 Dallas, USA), (…)” should be written:
“Linoleic acid (18:2ω6), linolenic acid (18:3ω3), AA and DHA (Santa Cruz Biotechnology, 105 Dallas, USA), (…)”
Please remember that your focus is on the nature of unsaturations, it is important for the reader to have a clear idea of what is ω6 or ω3.
These modifications have been made.
- In figure 2, please align “Muscle” with the other charts.
This has been corrected.
- Line 309
I would remove the “therefore” in the sentence. Please consider that you can have different FA assemblages with the same unsaturation rate. "therefore" implies that "various saturation rates" derive from the previous information, and you cannot conclude that. More, with your BD scheme, you mention unsaturation not saturation. Please correct.
This has been corrected.
- Line 317
Please don’t start a paragraph with “because”. You may give more emphasis to what muscle cells do as “Muscle cells undergo repeated cycles of elongation and relaxation, and are thus exposed to important mechanical constraints due to intrinsic repetitive muscle deformations (..)”
This has been corrected.
- Line 328-360
The authors give information that belongs to the methods. The authors should present the results and then reference the figures and supplement material. Even if the authors wish to present the method of cultivation here, they must present first the method and then the results. When describing the results, then they should refer the figures (Fig. 4, etc). The text must be revised here.
See preliminary comments. Since this paragraph has been amended according to Reviewer 1’s comments during the first round of review, we decided to keep it under its previous form.
- Figure 5. The information on the caption should be succinct. I would advise to transfer the mode of image capture to the methods section. The same applies to caption of figure 6 that I find unnecessarily long.
This has been corrected.
- Line 360
The authors mention “behaviors” of PL regarding the FA pattern of their acyl chains, which is not correct. The cells may have a behavioural pattern of incorporation, not the PLs. The sentence “Similar behaviors were observed with other PLs, namely PE, PS and PI (data not-shown).” when in fact what they observe as “The lack of AA and DHA incorporation into PC was also observed for the other PLs species PE, PS and PI (data not shown).”
The original sentence was changed to: “The same observation could be made for other PLs species PE, PS and PI (data not shown)”.
- Line 403-405 is just lost in the text. I don’t even know what the “therefore” means here. I suppose it follows the sentence ending in line 360, it should NOT be a different paragraph. I sound repetitive, but please ask yourselves what is the rule that you follow to make a new paragraph.
We agree with reviewer 1: this sentence was far from its original paragraph. The line break has been removed.
- Lines 406-417 are methods again. Some information is already given under “2.5. Cell labelling for confocal microscopy and methods for image analyses”, why not present the full description of methods in that section?
This has been corrected.
- As I mentioned in the previous revision, the authors should revise the text according to suggestions made in the text and apply those suggestion to the WHOLE text. It is not the job of the reviewer to edit the text sentence by sentence, the authors should do that. Take the example of sentence in line 440:
“Therefore, it appeared from these experiments that the cellular PL composition influences the ability of the cells to form VLDs under osmotic stress.” This is just messy, you should write something along the lines “These results suggest that the cellular PL composition influences the ability of the cells to form VLDs under osmotic stress”. I apologise for being blunt but “appeared from this experiments” is not a good way to convey the information that results point in a certain direction, it makes your results “appear” as casual observations, which they are not.
Please revise the remaining text, and check which information should be conveyed under “methods”.
Please see “Preliminary remarks”. The sentence has been modified as suggested by Reviewer 2.
